# PERRY: POLICY EVALUATION WITH CONFIDENCE INTERVALS USING AUXILIARY DATA

## ABSTRACT

Off-policy evaluation (OPE) methods estimate the value of a new reinforcement learning (RL) policy prior to deployment. Recent advances have shown that leveraging auxiliary datasets, such as those synthesized by generative models, can improve the accuracy of OPE. Unfortunately, such auxiliary datasets may also be biased, and existing methods for using data augmentation within OPE in RL lack principled uncertainty quantification. In high stakes settings like healthcare, reliable uncertainty estimates are important for ensuring safe and informed deployment. In this work, we propose two methods to construct valid confidence intervals for OPE when using data augmentation. The first provides a confidence interval over $V^\pi(s)$, the policy performance conditioned on an initial state $s$. To do so we introduce a new conformal prediction method suitable for Markov Decision Processes (MDPs) with high-dimensional state spaces. Second, we consider the more common task of estimating the average policy performance over many initial states, $V^\pi$; we introduce a method that draws on ideas from doubly robust estimation and prediction powered inference. Across simulators spanning robotics, healthcare and inventory management, and a real healthcare dataset from MIMIC-IV, we find that our methods can effectively leverage auxiliary data and consistently produce confidence intervals that cover the ground truth policy values, unlike previously proposed methods. Our work enables a future in which OPE can provide rigorous uncertainty estimates for high-stakes domains.

## 1 INTRODUCTION

Off-policy evaluation (OPE) (Precup et al., 2000; Sutton & Barto, 2018) is used to estimate the value of a new reinforcement learning (RL) policy prior to deployment using a historical dataset from a distinct behavior policy. This strategy is especially important in high-stakes domains (Gottesman et al., 2020; Mandel et al., 2014; Fu et al., 2020), where directly deploying new policies without prior evaluation can be costly or even harmful to participants. However, standard OPE methods frequently struggle when the target policy is very different than the behavior policy, due to limited dataset coverage (Jiang & Li, 2016). To address this, several recent works have proposed using synthetic auxiliary data to improve the coverage of the dataset and subsequently the accuracy of OPE methods (Tang & Wiens, 2023; Gao et al., 2024; Mandyam et al., 2024). Such approaches have either focused on the contextual bandit setting, or focused on promising empirical success in sequential settings but lack formal assurances on the quality of the proposed estimates.

However, in high stakes, multi-step domains, it is often of key importance to have confidence intervals (CIs) over the proposed policy estimates. Such intervals support safer, more informed policy selection and deployment. Therefore, we argue that principled uncertainty quantification is needed for OPE in RL in the emerging regime where both real and synthetic trajectories are used. While there is a notable body of prior work that developed CIs using *only* real data for OPE in RL (Thomas et al., 2015a;b; Taufiq et al., 2022; Foffano et al., 2023), to our knowledge, none provides guarantees in settings that combine offline and synthetic trajectories. In this paper we takes steps towards addressing this gap.

We formalize uncertainty quantification for OPE with mixed (real and synthetic) behavior data and identify two settings that require uncertainty-aware OPE. First, in domains like healthcare, it is common for stakeholders to deliberate between decision policies to use for individuals that start in

the same state: for example, a clinician may use the same treatment policy on all patients in the same stage of a disease. Estimating CIs for state-conditioned policy performance is thus an important task that can substantially benefit from data augmentation. Our first method, **CP-Gen**, provides conformal prediction intervals for such state-conditioned values. Second, we address evaluation of the target policy's expected value averaged over the distribution of initial states. We introduce a second method **DR-PPI**, which leverages techniques from doubly robust estimation and prediction-powered inference (Angelopoulos et al., 2023) to correct biases from generated trajectories and produce valid CIs.

Our empirical studies across inventory control, sepsis treatment, robotic control, and MIMIC-IV show that our methods, which can leverage synthetic data, can match or improve over state-of-the-art baselines that provide correct CIs using only real data. Our contributions follow.

1. **We formalize the problem of uncertainty quantification** for OPE in MDPs that leverage auxiliary data and introduce **CP-Gen** and **DR-PPI** (Section 3) for two natural settings where CIs are important.

2. **We prove that both methods yield valid CIs** and achieve the desired coverage probability either asymptotically or within a margin of error for finite sample sizes (Section 4).

3. **We empirically evaluate the estimators** in four domains including a real-world healthcare dataset, showing that our estimates which leverage auxiliary data produce CIs with the correct coverage that match or are tighter than baselines with valid coverage that do not use the auxiliary data. (Section 5).

## 2 BACKGROUND

### 2.1 PROBLEM SETTING

We consider a decision-making setting defined by the MDP $\mathcal{M} = (\mathcal{S}, \mathcal{A}, P, R, d_0, \gamma, H)$. $\mathcal{S}, \mathcal{A}$ denote the possibly infinite state and action spaces respectively. $P : \mathcal{S} \times \mathcal{A} \to \Delta(\mathcal{S})$ represents the transition dynamics, $R : \mathcal{S} \times \mathcal{A} \to \Delta(\mathbb{R})$ is the reward function, and $d_0 \in \Delta(\mathcal{S})$ is the initial state distribution. $\gamma$ is the discount factor and $H$ is the fixed horizon. A trajectory is defined as $\tau : \{s_t, a_t, r_t\}_{t=1}^{H}$ where $s_t, a_t, r_t$ are the corresponding state, action, and instantaneous reward observed at timestep $t$. The return of the trajectory $\tau$ is $J(\tau) = \sum_{t=1}^{H} \gamma^{t-1} r_t$ where $\pi$ is the policy used to generate the trajectory. The value of the policy $V^\pi = \mathbb{E}_{\tau \sim \pi}[J(\tau)]$ is calculated as an expectation over the possible trajectories that could arise from $\pi$. The value of a policy conditioned on an initial starting state $s$ is $V^\pi(s) = \mathbb{E}_{\tau \sim \pi}[J(\tau)|s_0 = s]$.

### 2.2 OFF-POLICY EVALUATION (OPE)

The goal of OPE is to estimate the value of a target policy $\pi_e$ given a dataset of behavior trajectories $D_{\pi_b}$ that arise from a distinct behavior policy $\pi_b$. In typical OPE setups, we assume access to $\pi_e$. In our work, we also assume $\pi_b$ is known, though we apply our methods empirically in settings where $\pi_b$ must be estimated. There are several standard approaches for OPE, including importance sampling (Precup et al., 2000), direct method (DM) (Li et al., 2010; Beygelzimer & Langford, 2009; van Seijen et al., 2009; Harutyunyan et al., 2016; Le et al., 2019; Voloshin et al., 2021), and doubly robust (DR) approaches (Farajtabar et al., 2018; Dudik et al., 2011; Jiang & Li, 2016). IS-based estimators re-weigh each trajectory in the $D_{\pi_b}$ using an inverse propensity score (IPS) $\rho(\tau) = \prod_{t=1}^{H} \frac{\pi_e(a_t|s_t)}{\pi_b(a_t|s_t)}$. DM estimators learn a reward model using the behavior trajectories to directly estimate the value of the target policy. DR methods combine the advantages of IS and DM estimators and provide favorable guarantees when either the IPS ratio or the reward model is inaccurate.

### 2.3 RELATED LITERATURE

**OPE with data augmentation**. As discussed in Section 1, standard OPE methods suffer when the behavior dataset has limited coverage. Because OPE methods typically assume finite sample sizes, OPE estimates can be either biased or have high variance (Precup et al., 2000; Jiang & Li,

2016; Thomas & Brunskill, 2016). To address this concern, several works have proposed using auxiliary information to enhance OPE estimators, using data augmentation either from a secondary dataset (Tang & Wiens, 2023; Mandyam et al., 2024) or by generating synthetic trajectories based on historical data (Gao et al., 2024; Sun et al., 2023; Gao et al., 2023). These works find that leveraging auxiliary data can substantially improve OPE estimates in some domains such as robotic control. However, these approaches may introduce additional bias due to errors in the auxiliary data, and lack theoretical guarantees or rigorous uncertainty quantification for the MDP setting.

**Conformal prediction for OPE**. Conformal prediction is a strategy to produce statistically valid prediction regions for any point prediction that arises from a machine learning (ML) model. Tibshirani et al. (2020) relaxed a limiting exchangeability assumption, and Taufiq et al. (2022) applied conformal prediction to the OPE setting for contextual bandits through reweighting. Foffano et al. (2023) later extended this work to create conformal intervals for OPE in MDPs. Crucially, their approximation relies on an integral that is difficult to compute and implement in continuous high-dimensional state spaces. Our `CP-Gen`, is inspired by the last approach, but uses a new way to compute the weights needed for the conformal prediction, allowing us to tackle settings with continuous and high-dimensional state spaces. In addition, this prior work did not consider data augmentation. Our new approach achieves tighter CIs through careful use of auxiliary datasets.

**Prediction-powered inference**. There are several challenges when applying ML methods to settings in which data is expensive to obtain. In these settings, it is useful to use ML models to generate predictions for unlabeled samples. Prediction-powered inference (PPI) allows us to calculate CIs on downstream task performance given both an original dataset and predictions from an ML model (Angelopoulos et al., 2023). PPI produces accurate CIs across a variety of ML tasks and dataset domains. PPI also has strong overlap with doubly robust estimation techniques typically used for OPE. However, the problem setup in PPI is distinct from ours. PPI assumes that we have access to a large dataset of observations that are unlabeled; the role of the ML model is to label the observations. In contrast, in our setting, we must both generate synthetic samples (i.e., trajectories) and their corresponding labels (i.e., returns); this necessitates a distinct methodology.

## 3 METHODS

In general, trajectories produced by generative models may be biased or drawn from a distribution distinct from that of the offline behavior policy, which can introduce error and/or variance into the resulting OPE estimate for sequential decision processes. We now describe two new methods for computing CIs for OPE in RL that use both offline and synthetically generated data for two common settings where CIs would be beneficial.

### 3.1 `CP-Gen`: CONFIDENCE INTERVALS FOR OPE FROM A STARTING STATE

Estimating state-conditioned policy values has been slightly understudied in the OPE for RL literature, which tends to focus on estimators that averaging performance over the full population of initial states. The task of estimating state-conditioned policy values can benefit substantially from data augmentation, since data from individual starting states is sparse, and yet obtaining valid CIs is of particular importance for high-stakes domains. To address this, we propose `CP-Gen`, a new conformal prediction method for OPE estimation for initial-state dependent policy value estimates.

Given an initial state $s$, we estimate $V^{\pi_e}(s)$ as follows:

$$
\begin{aligned}
V^{\pi_e}(s) &= \mathbb{E}_{\tau \sim p^{\pi_e}|s_0=s}[J(\tau)] \\
&= \sum_{\tau \sim p^{\pi_e}|s_0=s} p^{\pi_e}(\tau)J(\tau) \\
&= \sum_{\tilde{\tau} \sim \tilde{p}^{\pi_e}|s_0=s} \tilde{p}^{\pi_e}(\tilde{\tau})J(\tilde{\tau}) + \left[ \sum_{\tau \sim p^{\pi_e}|s_0=s} p^{\pi_e}(\tau)J(\tau) - \sum_{\tilde{\tau} \sim \tilde{p}^{\pi_e}|s_0=s} \tilde{p}^{\pi_e}(\tilde{\tau})J(\tilde{\tau}) \right] \\
&\approx \sum_{\tilde{\tau} \sim \tilde{p}^{\pi_e}|s_0=s} \tilde{p}^{\pi_e}(\tilde{\tau})J(\tilde{\tau}) + \frac{1}{n}\sum_{i=1}^{n} J(\tau_i|s_0=s) - \frac{1}{nM}\sum_{j=1}^{nM} J(\tilde{\tau}_j|s_0=s) \\
&= \sum_{\tilde{\tau} \sim \tilde{p}^{\pi_e}|s_0=s} \tilde{p}^{\pi_e}(\tilde{\tau})J(\tilde{\tau}) + \frac{1}{nM}\sum_{i=1}^{n}\sum_{j=1}^{M}(J(\tau_i|s_0=s) - J(\tilde{\tau}_{ij}|s_0=s)),
\end{aligned}
$$

where $\tilde{p}$ is the dynamics distribution induced by the generative model, $\tau \sim p^{\pi_e}$ is a trajectory drawn from the dynamics distribution associated with the policy $\pi_e$, $p^{\pi_e}(\tau)$ is the probability of observing trajectory $\tau$ under the policy $\pi_e$, $n/M$ is the number of behavior/synthetic trajectories, and $J(\tau|s_0=s)$ is the return of trajectory $\tau$ given initial state $s$.

Inspired by conformal prediction for regression, our goal is to produce an interval $\hat{C}_n(s)$ such that the return difference between any offline trajectory and its corresponding generated trajectory that starts from the same initial state $s$ lies in this band with high probability i.e.,

$$
P^{\pi_e}\left(J(\tau|s_0=s) - J(\tilde{\tau}|s_0=s) \in \hat{C}_n(s)\right) \geq \alpha, \tag{1}
$$

where $P^{\pi_e}$ is the probability measure induced by the target policy $\pi_e$ and $\alpha$ is the confidence level. Given this goal, the final conformal prediction interval for the value of the initial state $s$, $V^{\pi_e}(s)$, is

$$
\sum_{\tilde{\tau} \sim \tilde{p}^{\pi_e}, s_0=s} \tilde{p}^{\pi_e}(\tilde{\tau})J(\tilde{\tau}) + \hat{C}_n(s). \tag{2}
$$

If we set $\alpha$ appropriately (i.e., depending on the reward distribution), the band will cover the expected value of $R(\tau_i|s_0=s) - R(\tilde{\tau}_{ij}|s_0=s)$. This enables the final interval to cover $V^{\pi_e}(s)$.

---

**Algorithm 1 CP-Gen**

---

**Require:** Offline dataset $\mathcal{D}_{\pi_b}$, behavior policy $\pi_b$, target policy $\pi_e$, initial state $x$.
1: Split $D_{\pi_b}$ (size $K$) into $D_{tr}$ ($K/2$) and $D_{cal}$ ($K/2$)
2: Fit a generative model $\mathcal{T}$ using $D_{tr}$.
3: For each trajectory $\tau_i \in D_{tr}$, generate $M$ trajectories $\{\tilde{\tau}_{i,m}\}_{m=1}^{M}$ under $\pi_b$ with the same initial state as $\tau_i$, record the pairs as $\{(\tau_i, \tilde{\tau}_{i,m})\}_{m=1}^{M}$.
4: For each trajectory $\tau_j \in D_{cal}$, generate $N$ trajectories $\{\tilde{\tau}_{j,n}\}_{n=1}^{N}$ under $\pi_b$ with the same initial state as $\tau_j$, record the pairs as $\{(\tau_j, \tilde{\tau}_{j,n})\}_{n=1}^{N}$.
5: For each $(\tau_j, \tilde{\tau}_{j,n})$, calculate the weight $\hat{w}_\epsilon(x_j, J(\tau_j) - J(\tilde{\tau}_{j,n}))$ using $(\tau_i, \tilde{\tau}_{i,m})$ (Eqn (5)).
6: Given an initial state $x$, calculate $p_{j,n}^{\hat{w}}(x, y)$ and $p_{\frac{KN}{2}+1}^{\hat{w}}$ using Eqn (8).
7: For each $(\tau_j, \tilde{\tau}_{j,n})$, calculate the score $V_{j,n} = J(\tau_j) - J(\tilde{\tau}_{j,n})$.
8: Calculate $F^{(x,y)}$ using Eqn (7).
9: Calculate confidence interval $\hat{C}_{n,\alpha}$ over the value of trajectories starting in initial state $x$ using Eqn (6).
10: Rollout trajectories under $\pi_e$ from $\mathcal{T}$ and get the first term in Eqn (2).

---

To simplify notation, let $S$ be the initial state and $\Delta_{rr'}$ be the return difference of a pair of trajectories (one from the original behavior dataset, and one generated). Unlike standard conformal prediction, we must tackle the distribution shift induced by the difference between the behavior and target policies. To do so, prior work (Foffano et al., 2023), which builds on related work (Tibshirani et al., 2020; Taufiq et al., 2022), proposed CP methods for MDPs that weigh the calibration scores using estimates of the likelihood ratio.

However, this prior work does not consider the use of generated trajectories. Therefore we introduce a new sample reweighting technique that accounts for the distribution shift (from behavior to evaluation

policies) in both the real and generated trajectories (see full derivation in Appendix F):

$$w(s, \delta_{rr'}) := \mathbb{P}^{\pi_e}_{(S, \Delta_{rr'})}(s, \delta_{rr'}) / \mathbb{P}^{\pi_b}_{(S, \Delta_{rr'})}(s, \delta_{rr'}) \tag{3}$$

$$= \mathbb{E}_{\tau \sim p^{\pi_b}, \tilde{\tau} \sim \tilde{p}^{\pi_b}} \left[ \frac{\prod_{t=1}^{H} \pi_e(a_t|s_t)\pi_e(\tilde{a}_t|\tilde{s}_t)}{\prod_{t=1}^{H} \pi_b(a_t|s_t)\pi_b(\tilde{a}_t|\tilde{s}_t)} | s_0 = s, \delta_{J(\tau)J(\tilde{\tau})} = \delta_{rr'} \right]. \tag{4}$$

This weight is an expectation of the IPS ratio over all observations that share the same input ($s$) and score ($\delta_{rr'}$). However, calculating this will become intractable as the size of the MDP increases.

To mitigate this, and allow us to compute valid conformal prediction intervals in continuous state and action spaces, we use $\epsilon-$approximation to estimate the weight for a given sample:

$$w_\epsilon(s, \delta_{rr'}) = \mathbb{E}_{\tau \sim p^{\pi_b}, \tilde{\tau} \sim \tilde{p}^{\pi_b}} \left[ \frac{\prod_{t=1}^{H} \pi_e(a_t|s_t)\pi_e(\tilde{a}_t|\tilde{s}_t)}{\prod_{t=1}^{H} \pi_b(a_t|s_t)\pi_b(\tilde{a}_t|\tilde{s}_t)} | s_0 \in B(s, \epsilon_s), \delta_{J(\tau)J(\tilde{\tau})} \in B(\delta_{rr'}, \epsilon_r) \right],$$
$$\tag{5}$$

where $B(s, \epsilon_s)$ and $B(\delta_{rr'}, \epsilon_r)$ represent a ball around the input $s$ of radius $\epsilon_s$ and a ball around the output $\delta_{rr'}$ of radius $\epsilon_r$. This setup allows for small perturbations around $s$ and $\delta_{rr'}$. In particular, $B(s, \epsilon_s)$ captures any input $s$ that is within a small distance $\epsilon_s$ of $s$, and likewise for $B(\delta_{rr'}, \epsilon_r)$.

In this way, the weight $w_\epsilon(s, \delta_{rr'})$ is estimated using trajectories that are $\epsilon_s$-close in the initial state and $\epsilon_r$ close in the trajectory return (Algorithm 1). This approach allows us to calculate valid conformal prediction intervals, at the cost of a slight reduction in coverage. We will discuss this further in our theoretical analysis.

Using these weights, the CI band is as follows:

$$\hat{C}_{n,\alpha}(s) = \{\delta_{rr'} : Q(\frac{\alpha}{2}, F_n^{(s,\delta_{rr'})}) \le V_{n+1}^{(s,\delta_{rr'})} \le Q(1 - \frac{\alpha}{2}, F_n^{(s,\delta_{rr'})})\}, \tag{6}$$

where

$$F_n^{(s,\delta_{rr'})} = \sum_{i=1}^{n} p_i^w(s, \delta_{rr'})\delta_{V_i} + p_{n+1}^w(s, \delta_{rr'})\delta_\infty, \tag{7}$$

$$p_i^w(s, \delta_{rr'}) = \begin{cases} \frac{w(S_i, \Delta_{rr',i})}{\sum_{j=1}^{n} w(S_j, \Delta_{rr',j}) + w(s, \delta_{rr'})} & \text{if } i \le n, \\ \frac{w(s, \delta_{rr'})}{\sum_{j=1}^{n} w(S_j, \Delta_{rr',j}) + w(s, \delta_{rr'})} & \text{if } i = n+1, \end{cases} \tag{8}$$

$Q$ is a quantile, and $V_i = s(S_i, \Delta_{rr',i}) = \Delta_{rr',i}$. Typical conformal prediction methods do not provide coverage guarantees for individual samples. In our setting, however, the target of interest is $V^\pi(s)$, which is itself an expectation, so marginal coverage is sufficient.

---

**Algorithm 2 DR-PPI**

---

**Require:** Offline dataset $\mathcal{D}_{\pi_b}$, behavior policy $\pi_b$, target policy $\pi_e$.
  1: Split $D_{\pi_b}$ (size $n$) into $D_1$ and $D_2$ (each with size $\frac{n}{2}$).
  2: Fit a generative model $f_1$ using $D_1$.
  3: Use $f_1$ to generate $N_f$ rollouts $\{\tilde{\tau}_i\}_{i=1}^{N_f}$ from $\pi_e$.
  4: For each $\tau_j \in D_2$, use $f_1$ to generate $M$ rollouts $\{\tilde{\tau}_{m,j}\}_{m=1}^{M}$ with the same initial state $s_{0,j}$.
  5: Estimate $\hat{V}_{\text{DR-PPI:1}}$ using Eqn (9).
  6: Fit a generative model using $D_2$, and estimate $\hat{V}_{\text{DR-PPI:2}}$ in the same way.
  7: Estimate $\hat{V}^{\pi_e}$ using Eqn (18).
  8: Estimate the variance of $\hat{V}^{\pi_e}$ using Eqn (19).
  9: Provide confidence interval $\hat{C}_\alpha$ using Eqn (10).

---

### 3.2 DR-PPI: CONFIDENCE INTERVALS FOR UNCONDITIONAL OPE VALUE ESTIMATION

A more common task for OPE in RL is to estimate the value of the target policy averaged over initial states. This is relevant in settings where a single policy may be selected for the whole population, and a stakeholder wants to choose among different policies. Aggregating over the CP-Gen estimates using a union bound over initial states would make the CI estimates impractically wide. Instead we introduce a second estimator specifically for this setting, DR-PPI, which builds on prior literature

in doubly robust estimation and prediction-powered inference. Here, our goal is to construct an estimator of $V^{\pi_e} = \mathbb{E}_{s_0}[V^{\pi_e}(s_0)]$ with which we can calculate CIs.

First, we assume that the initial-state distribution $d_0$ is known (though our results extend to settings in which $d_0$ must be estimated). Now, we construct a cross-fitted, doubly-robust estimate of the policy value $V^{\pi_e}$ as follows. First, we split the behavior dataset $D_{\pi_b}$ into two equal parts, which we refer to as $D_1$ and $D_2$. We first use $D_1$ to fit a generative model $f_1$; this procedure is agnostic to the generative model used, and reasonable approaches include a diffusion model or a variational auto-encoder (VAE). Then, we use $f_1$ to generate $N_f$ rollouts $\{\tilde{\tau}_i\}_{i=1}^{N_f}$ where each rollout uses actions as sampled from the target policy $\pi_e$. The rollouts are used to calculate the model-based return; however since we expect this return to be biased, we add a correction term using the trajectories observed in $D_2$ as follows:

$$\widehat{V}_{\text{DR-PPI:1}}^{\pi_e} = \frac{1}{N_f} \sum_{i=1}^{N_f} J(\tilde{\tau}_i) \;+\; \frac{1}{n/2} \sum_{j \in D_2} \left( \tilde{J}(\tau_j) \;-\; \frac{1}{M} \sum_{m=1}^{M} J(\tilde{\tau}_{m,j} \mid s_{0,j}) \right), \tag{9}$$

where $n$ is the number original behavior trajectories, and $\tilde{J}(\tau_i)$ is the re-weighted return of the behavior trajectory $\tau_i$. We note that there are several possible ways to perform this re-weighting: IS, weighted IS (WIS), and per-decision IS (PDIS). Regardless of the re-weighting technique, our asymptotic theoretical results hold.

The importance-sampling based correction relies on the generation of an additional set of trajectories $\{\tilde{\tau}_{m,j}\}_{m=1}^{M}$, which all begin in the same initial state as the corresponding behavior trajectory $j$ but are generated from the target policy. The correction over $M$ generated trajectories does not need to be re-weighted because the trajectories are generated using $f_1$ from the target policy. To ensure that the data is used efficiently, we use cross-fitting (Chernozhukov et al., 2018) with two splits of the data. $\widehat{V}_{\text{DR-PPI:1}}$ uses $D_1$ to fit the generative model $f_1$ and uses $D_2$ to provide the correction. Similarly, we fit the generative model on $D_2$ to produce $f_2$ and correct the estimator using $D_1$, which yields $\widehat{V}_{\text{DR-PPI:2}}$. The final estimate (Algorithm 2) is the average of $\widehat{V}_{\text{DR-PPI:1}}$ and $\widehat{V}_{\text{DR-PPI:2}}$.

The variance can then be calculated by combining plug-in estimates of the variance of the model-based term and the IS-term for each dataset split (details in Appendix E). Using this variance, an approximate CI for a given choice of $(1 - \alpha)$ is

$$\widehat{C}_\alpha = \widehat{V}_{\text{DR-PPI}}^{\pi_e} \pm z_{1-\alpha/2} \sqrt{\mathbb{V}\left[\widehat{V}_{\text{DR-PPI}}^{\pi_e}\right]} \tag{10}$$

where $\mathbb{V}\left[\widehat{V}_{\text{DR-PPI}}^{\pi_e}\right]$ is the variance of the OPE estimate learned by DR-PPI.

Before analyzing the theoretical guarantees of the two estimators, we first compare their constructions. When all trajectories in an environment begin from the same initial state, the point estimates of both methods are identical, differing only in their confidence intervals. The re-weighting schemes, however, are distinct: DR-PPI re-weights only the real behavior trajectories, whereas CP-Gen uses a product of IPS ratios averaged across a set of trajectories. Finally, the return differences used to compute the CI in CP-Gen may exhibit higher variance than subtracting the mean of a set of trajectories from the return of a single trajectory in DR-PPI. However, this effect depends on the stochasticity of the generated trajectories and may vary across domains.

### 3.3 PRACTICAL CONSIDERATIONS

There are several practical considerations to enable OPE in environments with large state and action spaces as well as settings in which $\pi_b$ and $\pi_e$ differ substantially. First, it is occasionally necessary to clip the largest IPS ratios to avoid extremely large intervals. Ionides (2008) shows that using a clip constant set to $n^{1/2}$ where $n$ is the number of dataset samples, provides an optimal first order rate in the resulting mean-squared error of the OPE estimator, balancing the bias introduced by the clipping with the variance reduction. This clipping constant also ensures the resulting estimate is still consistent. Following this, we set the clipping constant at a rate of $n^{1/2}$.

Additionally, in Algorithm 2, we propose splitting the behavior dataset into two portions and aggregating the OPE estimate calculated using each portion. If a pre-trained generative model is available,

we use the full dataset to construct the CI, and no data splitting for generative model training is necessary. However, if no pre-trained model exists, we divide the data: one half is used to train the generative model, and the other half is used to compute the CI. Because these two subsets are independent, this preserves the exchangeability criterion for conformal prediction and the validity of `DR-PPI`. However, in practice, it may not possible to split the behavior dataset due to its size. For these settings, we choose not to perform cross-fitting, and instead report results without sub-dividing the dataset. As discussed in Section 5, this can still result in valid, but higher variance CIs.

Finally, for `CP-Gen`, we must set $\epsilon_s$ and $\epsilon_r$ depending on the environment. We view $\epsilon_s$ and $\epsilon_r$ as hyperparameters that need. One way to do this is via cross-validation, where we split the behavior dataset into training and validation sets, and choose the $\epsilon_r, \epsilon_s$ that yields the most accurate estimate of the value function $V^{\pi_b}$ on the validation set.

## 4 THEORETICAL RESULTS

Now, we discuss the theoretical guarantees of our approaches. As is standard in prior OPE literature, we assume that the target and behavior policies share common support, and that the instantaneous rewards and IPS ratios are bounded (Farajtabar et al., 2018; Thomas & Brunskill, 2016).

### 4.1 `CP-Gen` PRODUCES VALID CONFORMAL PREDICTION INTERVALS

We make a few additional assumptions to analyze `CP-Gen`. These assumptions balance theoretical rigor with practical relevance, allowing us to derive meaningful guarantees in high-dimensional, structured settings. Importantly, they still encompass a broad class of real-world MDPs.

**Assumption 1** (Lipschitz Continuity of the Policy). There exist constants $L_\pi, L_{\pi,s}, L_{\pi,a}$ such that for $\pi \in \{\pi_b, \pi_e\}$ and all $s, s_1 \in \mathcal{S}, a, a_1 \in \mathcal{A}$,

$$TV(\pi(\cdot|s), \pi(\cdot|s_1)) \leq L_\pi ||s - s_1|| \tag{11}$$

$$|\pi(a|s) - \pi(a_1|s_1)| \leq L_{\pi,s}||s - s_1|| + L_{\pi,a}||a - a_1||. \tag{12}$$

**Assumption 2** (Lipschitz Transition Dynamics). For all $s, s_1 \in \mathcal{S}, a, a_1 \in \mathcal{A}$,

$$TV(p(\cdot|s, a), p(\cdot|s_1, a_1)) \leq L_{p,s}||s - s_1|| + L_{p,a}||a - a_1||. \tag{13}$$

**Assumption 3** (Score Smoothness). The map $(s, \delta_{rr'}) \mapsto w(s, \delta_{rr'})$ is $L_r$-Lipschitz in its second argument: $|w(s, \delta_{rr'}) - w(s, \delta'_{rr'})| \leq L_r|\delta_{rr'} - \delta'_{rr'}|$.

We consider Assumptions 1 and 2 mild. In most cases, Assumption 1 holds with a sufficiently large Lipschitz constant; in practice, these constants are small when similar states are assigned similar actions, a condition often justified in domains like healthcare, where similar patients receive similar treatments. A comparable assumption has been studied in prior work (Liu et al., 2022). Similarly, Assumption 2 is a smoothness assumption on the transition dynamics which has been used in prior work (Asadi et al., 2018). For example, in healthcare, patients with comparable clinical profiles often respond similarly to similar treatments; small changes in dosage or patient characteristics typically produce gradual, not abrupt, differences in outcomes. Assumption 3 is perhaps the strongest and requires that the return differences between trajectories are smooth in their expected IPS ratios. However, in domains with a large number of samples, where we can use a more fine-grained $\epsilon_r$, the Lipschitz assumption here (which is multiplied by $\epsilon_r$ in our theoretical bound) will have much less impact. These assumptions are used to account for potential errors introduced by $\epsilon$-approximation, used in large or continuous state spaces and ensure that the resulting averaging error is bounded.

Under the stated assumptions, we now demonstrate that `CP-Gen` produces valid conformal prediction intervals within a small margin of error (Theorem 1, proof in Appendix F).

**Theorem 1** (Valid Conformal Prediction Interval). *Under Assumptions 1 to 3, suppose that $\mathbb{E}_{\pi_b}[|\hat{w}_\epsilon(S, \Delta_{rr'})|^k] \leq d^{2k}$ for some $k \geq 2$ and finite $d$. Then, define the estimation error $\Delta_w = \frac{1}{2}\mathbb{E}^{\pi_b}|\hat{w}_\epsilon(S, \Delta_{rr'}) - w(S, \Delta_{rr'})|$. We bound $\Delta_w$ as follows:*

$$\Delta_w = \tilde{\mathcal{O}}(n^{-1/2}\epsilon_s^{-3d_s/2}\epsilon_r^{-3/2} + \epsilon_s + \epsilon_r), \tag{14}$$

*where $d_s$ is the dimension of $\mathcal{S}$.*

*The coverage is then bounded as*

$$P^{\pi_e}(\Delta_{rr'} \in \hat{C}_{n,\alpha}(S)) \geq 1 - \alpha - \Delta_w. \tag{15}$$

*In addition, if the non-conformity scores $\{V_i\}_{i=1}^n$ have no ties almost surely, then*

$$P^{\pi_e}(\Delta_{rr'} \in \hat{C}_{n,\alpha}(S)) \leq 1 - \alpha - \Delta_w + cn^{1/k-1} \tag{16}$$

*for some positive constant $c$ depending on $d$ and $k$ only.*

Theorem 1 shows that $\epsilon$-approximation results in a loss of coverage specified by $\Delta_w$, which depends primarily on $\epsilon_s$ and $\epsilon_r$. In environments where these constants are small, or there are a large number of samples, or $\epsilon_s, \epsilon_r$ are optimally selected, we can get a smaller loss of coverage. We also note that the guarantee is similar in form to prior conformal intervals for MDPs (Foffano et al., 2023), but our construction has significant benefits over prior work: it can leverage synthetic data and allows us to compute CIs for high-dimensional, continuous states with our approximation of $w$.

### 4.2 `DR-PPI` PRODUCES ASYMPTOTICALLY VALID CONFIDNECE INTERVALS

In Section 3.2, we mention several choices for the importance-sampling correction including IS, WIS, and PDIS. Regardless of the correction style, we achieve asymptotically valid CIs (Theorem 2, proof in Appendix F).

**Theorem 2** (Asymptotically Valid CI). *For all possible corrections $\tilde{R}_{IS}$, $\tilde{R}_{WIS}$ and $\tilde{R}_{PDIS}$,*

$$\liminf_{n,M,N_f \to \infty} P(V^{\pi_e} \in \hat{C}_\alpha) \geq 1 - \alpha. \tag{17}$$

## 5 EMPIRICAL RESULTS

Our theoretical results demonstrated that `CP-Gen` and `DR-PPI` can calculate valid CIs under mild assumptions. To complement this analysis, we seek to answer the following questions using empirical results: **1)** Does the $\epsilon$-approximation used in `CP-Gen` cause the estimated interval to be biased? **2)** Do `DR-PPI` and `CP-Gen` produce intervals that cover the ground truth policy value? **3)** Under what conditions do the `DR-PPI` estimates outperform baseline approaches?

### 5.1 DATASETS

To answer our empirical questions, we use the following domains.

**Inventory Control** (Foffano et al., 2023): We adapt this simulator to accommodate a continuous state and reward space.

**Sepsis** (Oberst & Sontag, 2019): In this popular sepsis simulator, the goal is to successfully discharge a simulated patient. We approximate the dynamics using a feed-forward network.

**D4RL HalfCheetah** (Fu et al., 2020): The HalfCheetah environment is a Mujoco task in the D4RL suite where the goal is to get the cheetah to move forward. Here, we approximate the dynamics using a variational auto-encoder (VAE) (Gao et al., 2023).

**MIMIC-IV** (Johnson et al., 2020; Goldberger et al., 2000): We consider a subset of patients from MIMIC-IV that receive potassium repletion. To emulate a setting in which we have access to both a behavior and target cohort, we construct two sub-cohorts. The behavior sub-cohort consists of patients who receive lower dosages (<20 mEq/L), and the target sub-cohort consists of patients who receive higher dosages (>= 20mEq/L). We use a VAE to generate synthetic trajectories. Our goal is to learn the value of the target policy (i.e., repletion strategy in the higher-dosage cohort).

### 5.2 BASELINES

In addition to the baseline proposed in Foffano et al. (2023), we compare to the following approaches:

**Importance Sampling (IS)**: We use standard IS, deriving a bound using central limit theorem (CLT) or bootstrapping.

**Augmented Importance Sampling (AugIS)**: We use both the original dataset and a set of synthetic

trajectories to calculate an IS estimate, with bounds estimated using either CLT or boostrapping.

**Direct Method (DM)**: We use rollouts from the learned model and calculate the expectation of the trajectory returns. DM estimates do not produce CIs.

**Doubly Robust (DR)**: Here, we compute a DR estimated using DQL to learn the reward model.

**Augmented Doubly Robust (AugDR)**: Here, we use both offline trajectories and synthetic trajectories to learn a Deep Q-learning (DQL) reward model and then compute a DR estimate.

**Q-Bootstrap**: Here, we fit a $Q$-function using the behavior dataset and use it to learn a bootstrapped estimate of $V^{\pi_e}(s)$.

| Setting | $V^{\pi_e}(s)$ | DM | Foffano et al. | | Q-bootstrap | | CP-Gen | |
|---|---|---|---|---|---|---|---|---|
| | | | Interval | Covers? | Interval | Covers? | Interval | Covers? |
| Inventory | -412.85 | -120.57 | (-6040, 2510) | ✓ | (-1566.32, -1045.68) | ✗ | **(-4449.27, 1082.33)** | ✓ |
| Sepsis | -0.40 | -0.12 | **(-1,0)** | ✓ | (-0.01, 0.01) | ✗ | (-1.36, 0.54) | ✓ |
| D4RL Half Cheetah | 1990.39 | 1393.98 | (1750, 1940) | ✓ | (1820, 1880) | ✗ | **(1964.35, 2004.42)** | ✓ |
| MIMIC-IV | 1 | 0.689 | (0,1) | ✓ | (-1.28, 0.92) | ✓ | **(0.977, 1.1012)** | ✓ |

Table 1: **CP-Gen outperforms baselines across domains with continuous state-spaces**, producing conformal prediction intervals that cover the true policy value, $V^{\pi_e}(s)$. For methods that produce an interval, we report the interval for $\alpha = 0.05$ and whether the interval covers the true policy value. The method with the smallest interval length that covers the ground truth policy value is bolded.

| Setting | $V^{\pi_e}$ | IS (CLT) | IS (Bootstrap) | AugIS (CLT) | AugIS (Bootstrap) | DR (CLT) | AugDR (CLT) | DM | DR-PPI |
|---|---|---|---|---|---|---|---|---|---|
| Inventory | -428.51 | (-2139.27, -209.57) | (-2209.73, -227.59) | (-808.47, -753.81) | (-806.41, -756.73) | (-1804.16, 940.30) | (-914.72, -807.50) | -100.53 | **(-2106.27, -187.89)** |
| Sepsis | -0.56 | (-1.68, -0.10) | (-1.73, -0.25) | (-0.002, 0.006) | (-0.001, 0.006) | (-1.67, -0.44) | (-2.92e+10, 9.8e+10) | -0.4 | **(-1.45, -0.26)** |
| D4RL Half Cheetah | 1975.75 | (1814.37, 2096.25) | **(1802.37, 2074.04)** | (970.46, 1122.39) | (973.22, 1115.36) | (-1.320e+32, 4.194e+31) | (-3.59e+31, 7.58e+30) | 1423.57 | (1820.79, 2102.60) |
| MIMIC-IV | 0.746 | (0.31, 1.50) | **(0.56, 1.65)** | (0.711, 0.719) | (0.711, 0.718) | (-5.874e+21, 1.892e+21) | (0.719, 0.730) | 0.69 | (0.29, 1.48) |

Table 2: **DR-PPI produces valid confidence intervals across all domains.** We report all CIs for the same coverage ($\alpha = 0.05$), and bold the interval with the smallest size that also covers the ground truth policy value $V^{\pi_e}$.

### 5.3 RESULTS

**CP-Gen produces valid CP intervals**. As discussed in Section 3, to scale prior conformal prediction approaches to large MDPs, we use an $\epsilon$-approximation strategy. Despite this approximation, we find that **CP-Gen** still results in conformal prediction intervals that cover $V^{\pi_e}(s)$ with the specified confidence level, often with a smaller interval size than baseline approaches (Table 1). We compare to a DM-style baseline where we average the return of synthetic trajectories that start in the given initial state. We find that the DM baseline can produce a biased result with a poor generative model (e.g., in D4RL, MIMIC-IV). We also evaluate the baseline reported in Foffano et al. (2023). This baseline covers the ground truth value but produces wider intervals than our approach in all environments with continuous state spaces. These results suggest that **CP-Gen** newly enables conformal prediction for OPE in MDP settings with large state and action spaces.

Additionally, we discuss empirical coverage rates for the Inventory and Sepsis settings in Appendix A. We also note that in some experiments, we use behavior cloning to approximate $\pi_b$, and despite this approximation, find that our methods still produce valid CIs. This suggests that our approach can be practically robust to moderate misspecification of $\pi_b$. Finally, we study the effect of the number of synthetic trajectories and the quality of the synthetic trajectories on our proposed estimators in the Inventory and Sepsis settings. We find that our estimators improve as the number of generated trajectories increase and are robust to moderately noisy synthetic trajectories. Further details are in Appendix A.

**DR-PPI identifies valid confidence intervals that cover $V^{\pi_e}$ across all domains**. Across all domains, **DR-PPI** produces valid CIs that cover $V^{\pi_e}$, as our theoretical results suggest (Table 2). In

contrast, most baseline approaches either have wide CIs or have intervals that do not cover $V^{\pi_e}$. In fact, any baseline approach that uses generated trajectories (e.g., AugIS, AugDR) produces a biased interval, which suggests that naively incorporating auxiliary datasets results in a biased estimator. Furthermore, we find that in the D4RL and MIMIC-IV settings, DQL is unable to identify an accurate Q-learning function; as a result, the CIs become exponentially large.

**`DR-PPI` performs best in stochastic domains with high quality generative models**. Finally, we clarify the settings in which DR-PPI outperforms baselines. When the environment is deterministic (e.g., D4RL HalfCheetah), or the generative model is of poor quality (e.g., D4RL HalfCheetah, MIMIC-IV) `DR-PPI` performs similarly to the IS baselines. In such settings we do not get a favorable variance reduction from the synthetic trajectories. In contrast, in settings where the environment is stochastic and our learned generative model is good (e.g., Inventory, Sepsis), `DR-PPI` has tighter CIs. Given that both IS with bootstrapping and `DR-PPI` produce valid CIs, we recommend a simple rule: use the estimator with the narrower interval. We defer a rigorous selection criterion to future work.

## 6 CONCLUSION

Here, we take steps toward uncertainty-aware OPE in settings that combine real and synthetic trajectories. We present two complementary approaches, `CP-Gen` and `DR-PPI`, that use auxiliary data to construct CIs for OPE. `CP-Gen` calculates state-conditioned policy values, while `DR-PPI` estimates unconditional policy values. We provide theoretical guarantees (Section 4) and examine behavior across four domains, including a real-world EHR dataset (Section 5). Our results illustrate the feasibility of obtaining valid CIs with auxiliary data and highlight practical trade-offs to consider.

**Limitations and future work.** We work with two classes of generative models, VAEs and neural networks. Future work could explore alternatives such as diffusion models, develop principled procedures for setting $\epsilon_s$ and $\epsilon_r$, and investigate strategies to mitigate the impact of poor-quality generated trajectories. More broadly, we see value in analyzing these approaches under distribution shift and partial observability.

**Reproducibility statement.** The code to reproduce all experiments is anonymized here and will be released upon publication.

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

# A    ADDITIONAL EMPIRICAL RESULTS

First, we discuss additional empirical results. In the main text, we report **CP-Gen** and **DR-PPI** results across all domains. First, we include results in the Inventory setting that investigate the performance of our methods if the behavior policy $\pi_b$ is unknown. We report coverage rates across 50 trials, and define the estimation error in terms of $\epsilon_x$. Here, the true behavior policy is static and uniform across all actions $[1/11$ for - in range(11)]. The estimated behavior policy is defined as $[1/11 + \epsilon_x$ for - in range(5)] $+ [1/11 - \epsilon_x$ for - in range(5)] $+ [1/11]$. That is, the perturbation by increases the probability of the first 5 actions, decreases the probability of the next 5 actions, and retains the probability of the last action. Our results (Table 3) support the claim that our methods are robust to small errors in the estimation of the behavior policy, with coverage dropping at $\epsilon_x \geq 0.01$. We observe that the drop in coverage is more pronounced for **DR-PPI**, though we also note that an $\epsilon_x$ of 0.03 refers to a very large degree of misspecification.

Finally, we investigate empirical coverage rates for both methods in the Inventory and Sepsis settings (Table 4). We note that in both environments, **DR-PPI** covers the ground truth value of the policy, and that **CP-Gen** achieves the requested coverage in Inventory. We believe that the slight loss of coverage for **CP-Gen** in the Sepsis setting is due to a higher $\Delta_w$. In particular, the Sepsis environment, due to its discrete state and reward space, exhibits weak Lipschitz continuity, with a large Lipschitz constant. Furthermore, in this setting, $C_{ips}$, the upper bound of the IPS ratios, is large given that the target and behavior policies are quite distinct. As suggested in Theorem 1, these two factors contribute to a higher $\Delta_w$, which results in a small loss of coverage.

| Method | $\epsilon_x = 0$ | $\epsilon_x = 0.005$ | $\epsilon_x = 0.01$ | $\epsilon_x = 0.02$ | $\epsilon_x = 0.03$ |
|---|---|---|---|---|---|
| **CP-Gen** | 98% | 92% | 84% | 84% | 80% |
| **DR-PPI** | 96% | 94% | 90% | 82% | 72% |

Table 3: **CP-Gen and DR-PPI are robust to moderate levels of misspecification in** $\pi_b$**.** Here, we study the Inventory setting and report coverage rates (out of 50) as we vary $\epsilon_x$, the degree to which the behavior policy $\pi_b$ is perturbed. We request a coverage corresponding to $\alpha = 0.05$.

| Setting | Method | Coverage Rate | Average Length of Interval |
|---|---|---|---|
| Inventory | **CP-Gen** | 98% | 5576.85 |
| Inventory | **DR-PPI** | 96% | 1951.14 |
| Sepsis | **CP-Gen** | 92% | 1.442 |
| Sepsis | **DR-PPI** | 96% | 1.172 |

Table 4: **Empirical coverage rates across the Inventory and Sepsis settings for CP-Gen and DR-PPI.** We report coverage rates (out of 50 iterations) for corresponding to $\alpha = 0.05$.

Additionally, we discuss the performance of our estimators as a function of the number of generated synthetic trajectories and the noise of the synthetic trajectories (Figure 1). We find that in the majority of settings, **DR-PPI** and **CP-Gen** achieve the requested coverage ($\alpha = 0.05$). When **DR-PPI** does not achieve the requested coverage, we believe there are too few generated trajectories (i.e., non-asymptotic result). When **CP-Gen** does not achieve the requested coverage, we believe this is due to a higher $\Delta_w$ (similar to the empirical coverage experiment in Table 4). We also study the effect of the trajectory quality on the performance of our estimators. For the Inventory setting, we find that the coverage rate only slightly decreases at higher degrees of noise. For the sepsis setting, we find that coverage does not change in comparison to perfect generated trajectories, which suggests that our methods can correct for noisy synthetic trajectories in these settings.

# B    CODE

We include a Github link with our code, which we will make public upon acceptance. We also include code in our supplementary material.

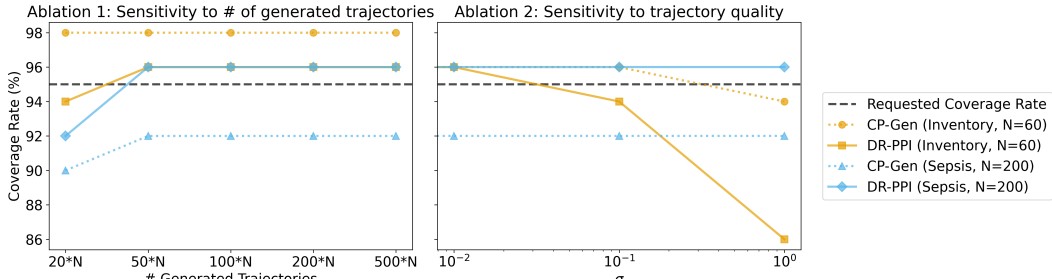

Figure 1: **DR-PPI** and **CP-Gen** are robust to annotation quality and improve in quality as the **number of generated trajectories increase** in the Inventory and Sepsis environments. (Left) We fix $N$ (i.e., number of behavior trajectories) for both the Inventory ($N = 60$) and Sepsis ($N = 200$) settings. We alter the number of generated trajectories from $20 * N$ to $500 * N$. We report the coverage rate across 50 iterations for $\alpha = 0.05$. (Right) We fix the number of generated trajectories at $100 * N$. We vary the quality of the generated trajectories by adding noise in the form of $\mathcal{N}(0, \sigma^2)$, similar to prior work (Laskin et al., 2020). We report coverage rate across 50 iterations, for $\alpha = 0.05$.

## C  EMPIRICAL SETTINGS

In the main text, we consider empirical results using four datasets. Here, we expand the description of each dataset.

**Inventory Control**:
The inventory control simulator is adapted from a version featured in Foffano et al. (2023). The state is the current inventory, actions are the number of units purchased, and reward is the end-of-day earnings. We make several adaptations to make this inventory control environment more suitable for our work. First, the distribution of the stochastic demand in the inventory is $o$ is changed from Poisson to normal $\mathcal{N}(\mu, \sigma)$. Additionally, the cost of buying items is $k \times \mathbb{1}_{\{a>0\}} + c(\min(N, x_t + a) - x_t)$, where $k > 0$ is the fixed cost for a single order, $c > 0$ is the cost of a single unit bought, $N$ is the inventory upper-bound, $x_t$ is the state at timestep $t$, and $a_t$ is the action at timestep $t$. The next state $x_{t+1}$ is calculated as $x_{t+1} = \max(0, \min(N, x_t + a_t) - o_{t+1})$. The instantaneous reward observed at the end of the day is the sum of the costs and earnings listed above (e.g., $r(x_t, a_t, x_{t+1}) = 10^2 \times (-k1_{\{a_t>0\}} - zx_t - c(\min(N, x_t + a_t) - x_t) + p\max(0, \min(N, x_t + a_t) - x_{t+1})))$. When testing our algorithm, we chose the following parameters: $N = 10, k = 1, c = 2, z = 2, p = 4, \mu = 5, \sigma = 10, H = 20$. We approximate the dynamics in this setting using a feed-forward network.

**Sepsis**:
The Sepsis simulator is taken directly from Oberst & Sontag (2019), which models a synthetic Sepsis treatment setting. The state is an 8-dimensional vector which contains information about relevant vitals and labs, indicators of ongoing treatment (e.g., antibiotics, vasopressors, ventilator), and an indication of whether the patient is diabetic. There are 8 possible actions, each corresponding to a different combination of 3 binary treatments (e.g., antibiotics, vasopressors, ventilator). The reward is $+1$ if the synthetic patient is off of treatment and has stable vitals, $-1$ if the patient has unstable vitals, and $+0$ otherwise. We do not alter any environment details, and report results with a maximum horizon of $H = 20$.

**D4RL HalfCheetah**:
The HalfCheetah environment is a Mujoco task in the D4RL suite (Fu et al., 2020). The cheetah is a two-dimensional robot that has 9 body parts and 8 joints connecting the body parts. Each state is represented as a 17-dimensional vector that contains information about the position and velocity of each of the joints. Each action is represented as a 5-dimensional vector, and applies torque to a subset of the joints, and the goal of the environment is to get the cheetah to move forward as quickly as possible. The reward corresponds to how far the cheetah traveled, with negative reward indicating that the cheetah moved backward. We report results using a maximum horizon of $H = 1000$.
**MIMIC-IV**:

MIMIC-IV is an electronic health records (EHR) dataset collected from patients admitted to the Beth Israel Deaconess Medical Center in Boston, MA (Johnson et al., 2020; Goldberger et al., 2000). We consider a subset of patients that receive potassium repletion through an intravenous (IV) line. We represent the patient state as a 20-dimensional vector containing information about important labs, administered medicines, and static covariates such as age and gender; each state represents a 4-hour interval in a patient's hospital stay. There are five actions, each corresponding to a dosage of potassium delivered through an IV. The reward function is a binary indicator of whether the patient's potassium lab value is within the potassium reference range (3.5-4.5 mmol/L), within 2 hours of receiving the administered potassium.

In the OPE task, we assume access to a behavior policy $\pi_b$ and a target policy $\pi_e$, and evaluate using RMSE. It is not immediately obvious how to create this setup within MIMIC-IV. To emulate a setting in which we have access to both a behavior and target cohort, we construct two sub-cohorts from the patients that receive potassium repletion. The behavior sub-cohort consists of patients who receive lower dosages (<20 mEq/L) of potassium, and the target sub-cohort consists of the patients who receive higher dosages (>= 20mEq/L) of potassium. The behavior policy corresponds to the repletion policy in the behavior cohort, and the target policy corresponds to the repletion policy in the target cohort. Both policies are inferred using behavior cloning. Our goal is to learn the value of the target policy, and a ground truth calculation of this value is calculated by averaging the returns of the target trajectories. We observe that the maximum horizon length is $H = 189$, though most patients have trajectories that are less than 20 timesteps.

## D    COMPUTATIONAL COST OF EXPERIMENTS

All experiments were conducted on an internally hosted cluster equipped with an NVIDIA RTX A6000 GPU featuring 48 GB of memory. In total, our experiments consumed approximately 250 compute hours, primarily driven by VAE training and Q-function learning on large datasets.

## E    DETAILS OF **DR-PPI** IMPLEMENTATION

As mentioned in Section 3.2, we use a cross-fitting technique to learn the final **DR-PPI** estimator. In particular, we average the outcomes of $\widehat{V}_{\texttt{DR-PPI}:1}$ and $\widehat{V}_{\texttt{DR-PPI}:2}$ as follows,

$$\widehat{V}_{\texttt{DR-PPI}} = (\widehat{V}_{\texttt{DR-PPI}:1} + \widehat{V}_{\texttt{DR-PPI}:2})/2. \tag{18}$$

The variance of the estimator can be calculated using plug-in estimates as follows,

$$\mathbb{V}\left[\widehat{V}_{\texttt{DR-PPI}}\right] = \frac{1}{4}\left(\frac{\widehat{\sigma}_{f_1}^2}{N_f} + \frac{\widehat{\sigma}_{b_1}^2}{n/2} + \frac{\widehat{\sigma}_{f_2}^2}{N_f} + \frac{\widehat{\sigma}_{b_2}^2}{n/2}\right), \tag{19}$$

where $\sigma_f^2 = \mathbb{V}_{\tilde{\tau} \sim \tilde{p}^{\pi_e}}[J(\tilde{\tau})]$, and $\sigma_b^2 = \mathbb{V}_{\tau \sim p^{\pi_b}, \tilde{\tau} \sim \tilde{p}^{\pi_e}}\left[\tilde{J}(\tau) - \frac{1}{M}\sum_{m=1}^{M} J(\tilde{\tau}_m | s_0(\tau))\right]$.

# F    PROOFS FOR THEORETICAL RESULTS

## F.1    PROOF OF EQN (4)

*Proof.*

$$w(s, \delta_{rr'}) := \frac{\mathbb{P}^{\pi_e}_{(S,\Delta_{rr'})}(s, \delta_{rr'})}{\mathbb{P}^{\pi_b}_{(S,\Delta_{rr'})}(s, \delta_{rr'})} \tag{20}$$

$$= \iint \frac{\mathbb{P}^{\pi_e}_{(S,\Delta_{rr'})}(s, \delta_{rr'})}{\mathbb{P}^{\pi_b}_{(S,\Delta_{rr'})}(s, \delta rr')} \frac{\mathbb{P}^{\pi_b}_{\tau,\tilde{\tau}|S,\Delta_{rr'}}(\tau, \tilde{\tau}|s, \delta_{rr'})}{\mathbb{P}^{\pi_b}_{\tau,\tilde{\tau}|S,\Delta_{rr'}}(\tau, \tilde{\tau}|s, \delta_{rr'})} \mathbb{P}^{\pi_e}_{\tau,\tilde{\tau}|S,\Delta_{rr'}}(\tau, \tilde{\tau}|s, \delta_{rr'}) d\tau d\tilde{\tau} \tag{21}$$

$$= \iint \frac{\mathbb{P}^{\pi_e}_{(S,\Delta_{rr'},\tau,\tilde{\tau})}(s, \delta_{rr'}, \tau, \tilde{\tau})}{\mathbb{P}^{\pi_b}_{(S,\Delta_{rr'},\tau,\tilde{\tau})}(s, \delta_{rr'}, \tau, \tilde{\tau})} \mathbb{P}^{\pi_b}_{\tau,\tilde{\tau}|S,\Delta_{rr'}}(\tau, \tilde{\tau}|s, \delta_{rr'}) d\tau d\tilde{\tau} \tag{22}$$

$$= \mathbb{E}_{\tau \sim p^{\pi_b}, \tilde{\tau} \sim \tilde{p}^{\pi_b}|S=s,\Delta_{rr'}=\delta_{rr'}}\left[\frac{\mathbb{P}^{\pi_e}_{(S,\Delta_{rr'},\tau,\tilde{\tau})}(s, \delta_{rr'}, \tau, \tilde{\tau})}{\mathbb{P}^{\pi_b}_{(S,\Delta_{rr'},\tau,\tilde{\tau})}(s, \delta_{rr'}, \tau, \tilde{\tau})}\right] \tag{23}$$

$$= \mathbb{E}_{\tau \sim p^{\pi_b}, \tilde{\tau} \sim \tilde{p}^{\pi_b}|S=s,\Delta_{rr'}=\delta_{rr'}}\left[\frac{P(\delta_{rr'}|s, \tau, \tilde{\tau})P^{\pi_e}(\tau|s)\tilde{P}^{\pi_e}(\tilde{\tau}|s)}{P(\delta_{rr'}|s, \tau, \tilde{\tau})P^{\pi_b}(\tau|s)\tilde{P}^{\pi_b}(\tilde{\tau}|s)}\right] \tag{24}$$

$$= \mathbb{E}_{\tau \sim p^{\pi_b}, \tilde{\tau} \sim \tilde{p}^{\pi_b}|S=s,\Delta_{rr'}=\delta_{rr'}}\left[\frac{P^{\pi_e}(\tau|s)\tilde{P}^{\pi_e}(\tilde{\tau}|s)}{P^{\pi_b}(\tau|s)\tilde{P}^{\pi_b}(\tilde{\tau}|s)}\right] \tag{25}$$

$$= \mathbb{E}_{\tau \sim p^{\pi_b}, \tilde{\tau} \sim \tilde{p}^{\pi_b}|S=s,\Delta_{rr'}=\delta_{rr'}}\left[\frac{\prod_{t=1}^{H} \pi_e(a_t|s_t)p(s_{t+1}|s_t, a_t)\pi_e(\tilde{a}_t|\tilde{s}_t)\tilde{p}(\tilde{s}_{t+1}|\tilde{s}_t, \tilde{a}_t)}{\prod_{t=1}^{H} \pi_b(a_t|s_t)p(s_{t+1}|s_t, a_t)\pi_b(\tilde{a}_t|\tilde{s}_t)\tilde{p}(\tilde{s}_{t+1}|\tilde{s}_t, \tilde{a}_t)}\right] \tag{26}$$

$$= \mathbb{E}_{\tau \sim p^{\pi_b}, \tilde{\tau} \sim \tilde{p}^{\pi_b}|S=s,\Delta_{rr'}=\delta_{rr'}}\left[\frac{\prod_{t=1}^{H} \pi_e(a_t|s_t)\pi_e(\tilde{a}_t|\tilde{s}_t)}{\prod_{t=1}^{H} \pi_b(a_t|s_t)\pi_b(\tilde{a}_t|\tilde{s}_t)}\right] \tag{27}$$

□

## F.2    ADDITIONAL ASSUMPTIONS

First, we formally state the assumptions used in prior literature (Farajtabar et al., 2018; Thomas & Brunskill, 2016) to support our theoretical results.

**Assumption 4** (Common support). $\pi_e(a|s) > 0 \rightarrow \pi_b(a|s) > 0, \forall s \in \mathcal{S}, \forall a \in \mathcal{A}$.

**Assumption 5** (Bounded return). $0 \leq J(\tau) \leq C_r$ for all $\tau \sim p$.

**Assumption 6** (Bounded IPS weights). $c_{ips} \leq \frac{\pi_e(a|s)}{\pi_b(a|s)} \leq C_{ips}, \forall s \in \mathcal{S}, \forall a \in \mathcal{A}$.

These assumptions are standard in the literature and minimally restrictive, thus enabling the analysis of `CP-Gen`'s performance under realistic conditions. We also consider Assumption 7, a mild regularity condition that holds in a wide variety of real-world MDPs, including those with heterogeneous populations and varied outcomes, such as clinical settings with diverse patient cohorts. Prior work in bandit and reinforcement learning has used similar assumptions (Qian & Yang, 2016; Bastani et al., 2020; Neu et al., 2010).

**Assumption 7** (Bounded density). The joint density of $(S, \Delta_{rr'})$ under $\mathbb{P}^{\pi_b}$ is uniformly bounded: $p_{\min} \leq p(s, \delta_{rr'}) \leq p_{\max}, \forall s, \delta_{rr'}$.

## F.3    PROOF OF THEOREM 1

**Lemma 3** (Coupling Lemma). *Let $X$ and $Y$ be random variables with probability distributions $\mu$ and $\nu$ over $\Omega$. There always exists a coupling $w$ on $\Omega \times \Omega$ s.t.,*

$$\|\mu - \nu\|_{TV} = P(X \neq Y).$$

*Proof.* This is a prior known result. Reference includes Daskalakis et al. (2011).    □

**Lemma 4.** *Assume the action space is bounded, $\|a\| \leq C_a$. Given two states $s, s_1$, there exists an optimal coupling, such that*

$$\mathbb{E}_{a \sim \pi(\cdot|s), a_1 \sim \pi(\cdot|s_1)} \|a - a_1\| \leq 2C_a P^\pi(a \neq a_1) = 2C_a TV(\pi(\cdot|s), \pi(\cdot|s_1)) \leq 2C_a L_\pi \|s - s_1\|. \tag{28}$$

*Proof.* This is a direct consequence of Coupling Lemma. $\square$

**Lemma 5.** *Assume the action space is bounded, $\|s\| \leq C_s$. Given two states $s_{t-1}, s'_{t-1}$, there exists an optimal coupling, such that*

$$\mathbb{E}_{a_{t-1} \sim \pi(\cdot|s_{t-1}), a'_{t-1} \sim \pi(\cdot|s'_{t-1}), s_t \sim p(\cdot|s_{t-1}, a_{t-1}), s'_t \sim p(\cdot|s'_{t-1}, a'_{t-1})} \|s_t - s'_t\| \tag{29}$$

$$\leq 2C_s P^\pi(s_t \neq s'_t) \tag{30}$$

$$= 2C_s \mathbb{E}_{a_{t-1} \sim \pi(\cdot|s_{t-1}), a'_{t-1} \sim \pi(\cdot|s'_{t-1})} TV(p(\cdot|s_{t-1}, a_{t-1}), p(\cdot|s'_{t-1}, a'_{t-1})) \tag{31}$$

$$\leq 2C_s(L_{p,s}\|s_{t-1} - s'_{t-1}\| \tag{32}$$

$$+ L_{p,a} \mathbb{E}_{a_{t-1} \sim \pi(\cdot|s_{t-1}), a'_{t-1} \sim \pi(\cdot|s'_{t-1})} \|a_{t-1} - a'_{t-1}\|) \tag{33}$$

$$\leq 2C_s(L_{p,s} + 2C_a L_\pi L_{p,a})\|s_{t-1} - s'_{t-1}\|. \tag{34}$$

*Thus, if $\|s_1 - s'_1\| \leq \epsilon_s$, then*

$$\mathbb{E}_{\tau, \tau' \sim p^\pi} \|s_t - s'_t\| \leq L^{t-1} \mathbb{E}_{\tau, \tau' \sim p^\pi} \|s_1 - s'_1\| \leq \epsilon_s L^{t-1}, \tag{35}$$

*where $L = 2C_s(L_{p,s} + 2C_a L_\pi L_{p,a})$.*

*And the same holds also for $\tilde{p}$.*

*Proof.* This is a direct consequence of Coupling Lemma. $\square$

**Lemma 6.** *$\forall s, a, s', a', s_1, a_1, s'_1, a'_1$, for $\pi \in \{\pi_b, \pi_e\}$,*

$$|\pi(a|s)\pi(a'|s') - \pi(a_1|s_1)\pi(a'_1|s'_1)| \leq L_{\pi,s}(\|s - s_1\| + \|s' - s'_1\|) + L_{\pi,a}(\|a - a_1\| + \|a' - a'_1\|). \tag{36}$$

*Proof.*

$$|\pi(a|s)\pi(a'|s') - \pi(a_1|s_1)\pi(a'_1|s'_1)| \tag{37}$$

$$\leq |\pi(a|s)\pi(a'|s') - \pi(a|s)\pi(a'_1|s'_1)| + |\pi(a|s)\pi(a'_1|s'_1) - \pi(a_1|s_1)\pi(a'_1|s'_1)| \tag{38}$$

$$\leq L_{\pi,s}\|s' - s'_1\| + L_{\pi,a}\|a' - a'_1\| + L_{\pi,s}\|s - s_1\| + L_{\pi,a}\|a - a_1\|. \tag{39}$$

$\square$

**Lemma 7.** *Assume $\forall s, a$, for $\pi \in \{\pi_b, \pi_e\}$, $\pi(a|s) \geq c > 0$. Define the per-step importance-ratio*

$$f(s, a, s', a') = \frac{\pi_e(a|s)\pi_e(a'|s')}{\pi_b(a|s)\pi_b(a'|s')},$$

*we can derive that there is a constant $L_f(c, c_{ips}, C_{ips}, L_{\pi,s}, L_{\pi,a})$ such that*

$$|f(s, a, s', a') - f(s_1, a_1, s'_1, a'_1)| \leq L_f(\|s - s_1\| + \|s' - s'_1\| + \|a - a_1\| + \|a' - a'_1\|). \tag{40}$$

*Proof.*

$$|f(s, a, s', a') - f(s_1, a_1, s'_1, a'_1)| = \frac{|\pi_e(a|s)\pi_e(a'|s')\pi_b(a_1|s_1)\pi_b(a'_1|s'_1) - \pi_e(a_1|s_1)\pi_e(a_1|s'_1)\pi_b(a|s)\pi_b(a'|s')|}{\pi_b(a|s)\pi_b(a'|s')\pi_b(a_1|s_1)\pi_b(a'_1|s'_1)} \tag{41}$$

$$\leq \frac{|\pi_e(a|s)\pi_e(a'|s')\pi_b(a_1|s_1)\pi_b(a'_1|s'_1) - \pi_e(a|s)\pi_e(a'|s')\pi_b(a|s)\pi_b(a'|s')|}{\pi_b(a|s)\pi_b(a'|s')\pi_b(a_1|s_1)\pi_b(a'_1|s'_1)} \tag{42}$$

$$+ \frac{|\pi_e(a|s)\pi_e(a'|s')\pi_b(a|s)\pi_b(a'|s') - \pi_e(a_1|s_1)\pi_e(a_1|s'_1)\pi_b(a|s)\pi_b(a'|s')|}{\pi_b(a|s)\pi_b(a'|s')\pi_b(a_1|s_1)\pi_b(a'_1|s'_1)} \tag{43}$$

$$\leq 2c^4(L_{\pi,s}\|s' - s'_1\| + L_{\pi,a}\|a' - a'_1\| + L_{\pi,s}\|s - s_1\| + L_{\pi,a}\|a - a_1\|) \tag{44}$$

$$\leq L_f(\|s - s_1\| + \|s' - s'_1\| + \|a - a_1\| + \|a' - a'_1\|). \tag{45}$$

$\square$

**Theorem 8** ($\epsilon-$approximation Error Bound).

$$|w_\epsilon(s, \delta_{rr'}) - w(s, \delta_{rr'})| \le L_s \epsilon_s + L_r \epsilon_r,$$

*where*

$$L_s = 2C_{ips}^{2(H-1)}(2C_a L_\pi + 1)L_f \frac{L^H - 1}{L - 1}$$

*Proof.* Define

$$g(\tau, \tau') = \prod_{t=1}^{H} f(s_t, a_t, s'_t, a'_t).$$

By the telescoping-product bound and the Lipschitz of each $f$,

$$|g(\tau, \tau') - g(\tau_1, \tau'_1)| = |\prod_{t=1}^{H} f(s_t, a_t, s'_t, a'_t) - \prod_{t=1}^{H} f(s_{1,t}, a_{1,t}, s'_{1,t}, a'_{1,t})| \tag{46}$$

$$= |\prod_{t=1}^{H} f(s_t, a_t, s'_t, a'_t) - \prod_{t=1}^{H-1} f(s_t, a_t, s'_t, a'_t)f(s_{1,H}, a_{1,H}, s'_{1,H}, a'_{1,H}) \tag{47}$$

$$+ \prod_{t=1}^{H-1} f(s_t, a_t, s'_t, a'_t)f(s_{1,H}, a_{1,H}, s'_{1,H}, a'_{1,H}) - \prod_{t=1}^{H-2} f(s_t, a_t, s'_t, a'_t) \prod_{t=H-1}^{H} f(s_{1,t}, a_{1,t}, s'_{1,t}, a'_{1,t}) \tag{48}$$

$$+ \cdots - \prod_{t=1}^{H} f(s_{1,t}, a_{1,t}, s'_{1,t}, a'_{1,t})| \tag{49}$$

$$\le \sum_{t=1}^{H} (\prod_{i \ne t} C_{ips}^2)|f(s_t, a_t, s'_t, a'_t) - f(s_{1,t}, a_{1,t}, s'_{1,t}, a'_{1,t})| \tag{50}$$

$$\le C_{ips}^{2(H-1)} L_f \sum_{t=1}^{H} (\|s_t - s_{1,t}\| + \|a_t - a_{1,t}\| + \|s'_t - s'_{1,t}\| + \|a'_t - a'_{1,t}\|). \tag{51}$$

Taking expectations under the optimal coupling gives

$$|w(s, \delta_{rr'}) - w(s', \delta_{rr'})| = |\mathbb{E}_{\tau \sim p^{\pi_b}, \tau' \sim \tilde{p}^{\pi_b}}[g(\tau, \tau') \mid s] - \mathbb{E}_{\tau_1 \sim p^{\pi_b}, \tau'_1 \sim \tilde{p}^{\pi_b}}[g(\tau_1, \tau'_1) \mid s']| \tag{52}$$

$$\le \mathbb{E}_{\tau, \tau_1 \sim p^{\pi_b}, \tau', \tau'_1 \sim \tilde{p}^{\pi_b}}[|g(\tau, \tau') - g(\tau_1, \tau'_1)| \mid s, s'] \tag{53}$$

$$= C_{ips}^{2(H-1)} L_f \sum_{t=1}^{H} \mathbb{E}_{\tau, \tau_1 \sim p^{\pi_b}, \tau', \tau'_1 \sim \tilde{p}^{\pi_b}}(\|s_t - s_{1,t}\| + \|a_t - a_{1,t}\| + \|s'_t - s'_{1,t}\| + \|a'_t - a'_{1,t}\|) \tag{54}$$

$$\le C_{ips}^{2(H-1)} L_f \sum_{t=1}^{H} 2(2C_a L_\pi + 1)\epsilon_s L^{t-1} \tag{55}$$

$$= L_s \epsilon_s \tag{56}$$

Hence $x \mapsto w(x, y)$ is $L_s$-Lipschitz. Finally, notice that

$$w_\epsilon(s, \delta_{rr'}) = \mathbb{E}^{\pi_b}\big[w(S, \Delta_{rr'}) \mid S \in B(s, \epsilon_s), \Delta_{rr'} \in B(\delta_{rr'}, \epsilon_r)\big],$$

so

$$\big|w_\epsilon(s, \delta_{rr'}) - w(s, \delta_{rr'})\big| = \mathbb{E}^{\pi_b}\big[w(S, \Delta_{rr'}) - w(s, \delta_{rr'}) \mid S \in B(s, \epsilon_s), \Delta_{rr'} \in B(\delta_{rr'}, \epsilon_r)\big] \tag{57}$$

$$\le \sup_{\|s'-s\| \le \epsilon_s, |\delta'_{rr'} - \delta_{rr'}| \le \epsilon_r} \big|w(s', \delta'_{rr'}) - w(s, \delta_{rr'})\big| \tag{58}$$

$$\le L_s \epsilon_s + L_r \epsilon_r. \tag{59}$$

as claimed. □

**Lemma 9.** *There exists an* $(\epsilon_s^0, \epsilon_r^0)-$*covering of* $S \times \Delta_{rr'}$, *denoted as* $\mathcal{N} = \{(s_i, \delta_{rr',j})\}_{i=1,\dots,N_S(\epsilon_s^0); j=1,\dots,N_{\Delta_{rr'}}(\epsilon_r^0)}$, *such that with probability* $\geq 1 - \delta$,

$$|\hat{w}_\epsilon(s_i, \delta_{rr',j}) - w_\epsilon(s_i, \delta_{rr',j})| \leq (C_{ips}^2 - c_{ips}^2)\sqrt{\frac{\ln(2N_S(\epsilon_s^0)N_{\Delta_{rr'}}(\epsilon_r^0)/\delta)}{N_{\min}}}, \forall i, j, \quad (60)$$

*where* $N_{\min} = np_{\min}Vol(B(\epsilon_s, \epsilon_r))$.

We now start the main proof of Theorem 1.

*Proof.*

$$\mathbb{E}^{\pi_b}|\hat{w}_\epsilon(S, \Delta_{rr'}) - w(S, \Delta_{rr'})| \leq \mathbb{E}^{\pi_b}|\hat{w}_\epsilon(S, \Delta_{rr'}) - w_\epsilon(S, \Delta_{rr'})| + \mathbb{E}^{\pi_b}|w_\epsilon(S, \Delta_{rr'}) - w(S, \Delta_{rr'})|$$
$$(61)$$

$$\leq \mathbb{E}^{\pi_b}|\hat{w}_\epsilon(S, \Delta_{rr'}) - w_\epsilon(S, \Delta_{rr'})| + L_s\epsilon_s + L_r\epsilon_r \quad (62)$$

For each $(s, \delta_{rr'})$, $\exists(s_i, \delta_{rr',j}) \in \mathcal{N}$, such that $\|s - s_i\| \leq \epsilon_s^0, \|\delta_{rr'} - \delta_{rr',j}\| \leq \epsilon_r^0$, so

$$\mathbb{E}^{\pi_b}|\hat{w}_\epsilon(S, \Delta_{rr'}) - w_\epsilon(S, \Delta_{rr'})| \leq \mathbb{E}^{\pi_b}[\underbrace{|\hat{w}_\epsilon(S, \Delta_{rr'}) - \hat{w}_\epsilon(s_i, \delta_{rr',j})|}_{1} + \underbrace{|\hat{w}_\epsilon(s_i, \delta_{rr',j}) - w_\epsilon(s_i, \delta_{rr',j})|}_{2}$$
$$(63)$$

$$+ \underbrace{|w_\epsilon(s_i, \delta_{rr',j}) - w_\epsilon(S, \Delta_{rr'})|}_{3}] \quad (64)$$

Bounding (1) by Assumption 7:

$$|\hat{w}_\epsilon(s, \delta_{rr'}) - \hat{w}_\epsilon(s_i, \delta_{rr',j})| \quad (65)$$

$$= |\frac{1}{N(s, \delta_{rr'}, \epsilon_s, \epsilon_r)} \sum_{(k,k') \in N(s, \delta_{rr'}, \epsilon_s, \epsilon_r)} \frac{\prod_{t=1}^H \pi_e(a_t^k|s_t^k)\pi_e(a_t^{k'}|s_t^{k'})}{\prod_{t=1}^H \pi_b(a_t^k|s_t^k)\pi_b(a_t^{k'}|s_t^{k'})} \quad (66)$$

$$- \frac{1}{N(s_i, \delta_{rr',j}, \epsilon_s, \epsilon_r)} \sum_{(k,k') \in N(s_i, \delta_{rr',j}, \epsilon_s, \epsilon_r)} \frac{\prod_{t=1}^H \pi_e(a_t^k|s_t^k)\pi_e(a_t^{k'}|s_t^{k'})}{\prod_{t=1}^H \pi_b(a_t^k|s_t^k)\pi_b(a_t^{k'}|s_t^{k'})}| \quad (67)$$

$$\leq |\frac{1}{N(s, \delta_{rr'}, \epsilon_s, \epsilon_r)}(\sum_{(k,k') \in N(s, \delta_{rr'}, \epsilon_s, \epsilon_r)} \frac{\prod_{t=1}^H \pi_e(a_t^k|s_t^k)\pi_e(a_t^{k'}|s_t^{k'})}{\prod_{t=1}^H \pi_b(a_t^k|s_t^k)\pi_b(a_t^{k'}|s_t^{k'})} - \sum_{(k,k') \in N(s_i, \delta_{rr',j}, \epsilon_s, \epsilon_r)} \frac{\prod_{t=1}^H \pi_e(a_t^k|s_t^k)\pi_e(a_t^{k'}|s_t^{k'})}{\prod_{t=1}^H \pi_b(a_t^k|s_t^k)\pi_b(a_t^{k'}|s_t^{k'})})|$$
$$(68)$$

$$+ |(\frac{1}{N(s, \delta_{rr'}, \epsilon_s, \epsilon_r)} - \frac{1}{N(s_i, \delta_{rr',j}, \epsilon_s, \epsilon_r)}) \sum_{(k,k') \in N(s_i, \delta_{rr',j}, \epsilon_s, \epsilon_r)} \frac{\prod_{t=1}^H \pi_e(a_t^k|s_t^k)\pi_e(a_t^{k'}|s_t^{k'})}{\prod_{t=1}^H \pi_b(a_t^k|s_t^k)\pi_b(a_t^{k'}|s_t^{k'})}|]$$
$$(69)$$

$$\leq 2d^{2H}\frac{p_{\max}}{p_{\min}}\frac{Vol(\text{Diff}(B(s, \delta_{rr'}, \epsilon_s, \epsilon_r), B(s_i, \delta_{rr',j}, \epsilon_s, \epsilon_r)))}{Vol(B(\epsilon_s, \epsilon_r))} \quad (70)$$

$$= \tilde{\mathcal{O}}(\frac{2d^{2H}p_{\max}\epsilon_s^0\epsilon_s^{d_s-1}\epsilon_r^0}{p_{\min}\epsilon_s^{d_s}\epsilon_r}), \quad (71)$$

where the last equation is followed by Li (2011).

Bounding (2) by Lemma 9:

Because with probability $\geq 1 - \delta$,

$$|\hat{w}_\epsilon(s_i, \delta_{rr',j}) - w_\epsilon(s_i, \delta_{rr',j})| \leq (C_{ips}^2 - c_{ips}^2)\sqrt{\frac{\ln(2N_S(\epsilon_s^0)N_{\Delta_{rr'}}(\epsilon_r^0)/\delta)}{N_{\min}}}, \forall i, j, \quad (72)$$

let $t = (C_{ips}^2 - c_{ips}^2)\sqrt{\frac{\ln(2N_S(\epsilon_s^0)N_{\Delta_{rr'}}(\epsilon_r^0)/\delta)}{N_{\min}}}$, we have $\delta = 2N_S(\epsilon_s^0)N_{\Delta_{rr'}}(\epsilon_r^0)e^{-(\frac{t}{C_{ips}^2 - c_{ips}^2})^2 N_{\min}}$, so

$$P^{\pi_b}(|\hat{w}_\epsilon(s_i, \delta_{rr',j}) - w_\epsilon(s_i, \delta_{rr',j})| \geq t) \leq 2N_S(\epsilon_s^0)N_{\Delta_{rr'}}(\epsilon_r^0)e^{-(\frac{t}{C_{ips}^2 - c_{ips}^2})^2 N_{\min}}.$$

$$\mathbb{E}^{\pi_b}[|\hat{w}_\epsilon(s_i, \delta_{rr',j}) - w_\epsilon(s_i, \delta_{rr',j})|] = \int_0^\infty P^{\pi_b}(|\hat{w}_\epsilon(s_i, \delta_{rr',j}) - w_\epsilon(s_i, \delta_{rr',j})| \geq t)dt \tag{73}$$

$$\leq \int_0^\infty 2N_S(\epsilon_s^0)N_{\Delta_{rr'}}(\epsilon_r^0)e^{-(\frac{t}{C_{ips}^2 - c_{ips}^2})^2 N_{\min}}dt \tag{74}$$

$$= \frac{(C_{ips}^2 - c_{ips}^2)N_S(\epsilon_s^0)N_{\Delta_{rr'}}(\epsilon_r^0)\sqrt{\pi}}{\sqrt{N_{\min}}} \tag{75}$$

$$= \tilde{\mathcal{O}}\left(\frac{(1 + \frac{1}{\epsilon_s^0})^{d_s}(1 + \frac{1}{\epsilon_r^0})}{\sqrt{np_{\min}\epsilon_s^{d_s}\epsilon_r}}\right) \tag{76}$$

Bounding (3) by Lipschitz property:

$$\mathbb{E}^{\pi_b}[|w_\epsilon(s_i, \delta_{rr',j}) - w_\epsilon(S, \Delta_{rr'})|] \leq L_s\epsilon_s + L_r\epsilon_r. \tag{77}$$

Putting it all together,

$$\mathbb{E}^{\pi_b}|\hat{w}_\epsilon(S, \Delta_{rr'}) - w(S, \Delta_{rr'})| = \tilde{\mathcal{O}}\left(\frac{2d^{2H}p_{\max}\epsilon_s^0\epsilon_s^{d_s-1}\epsilon_r^0}{p_{\min}\epsilon_s^{d_s}\epsilon_r} + \frac{(1 + \frac{1}{\epsilon_s^0})^{d_s}(1 + \frac{1}{\epsilon_r^0})}{\sqrt{np_{\min}\epsilon_s^{d_s}\epsilon_r}} + \epsilon_s + \epsilon_r\right)$$

$$\tag{78}$$

$$= \tilde{\mathcal{O}}(n^{-1/2}\epsilon_s^{-3d_s/2}\epsilon_r^{-3/2} + \epsilon_s + \epsilon_r), \tag{79}$$

where the last step follows by setting $\epsilon_s^0 = \epsilon_s, \epsilon_r^0 = \epsilon_r$.

The rest of the proof follows directly from Proposition 2 in (Foffano et al., 2023). $\square$

### F.4 PROOF OF THEOREM 2

We use Assumptions 4 to 6, which are standard in prior OPE literature.

*Proof.* Because

$$\mathbb{E}_{\pi_b}[\tilde{J}_{\text{IS}}(\tau_i)] = \mathbb{E}_{\pi_b}\left[\frac{\prod_{t=1}^H \pi_e(a_t^i|s_t^i)}{\prod_{t=1}^H \pi_b(a_t^i|s_t^i)}J(\tau_i)\right] = V^{\pi_e}, \tag{80}$$

and

$$\mathbb{E}_{\pi_b}[\tilde{J}_{\text{PDIS}}(\tau_i)] = \mathbb{E}^{\pi_b}\left[\sum_{t=1}^H \gamma^{t-1}\prod_{k=1}^t \frac{\pi_e(a_k^i \mid s_k^i)}{\pi_b(a_k^i \mid s_k^i)}r_t\right] = V^{\pi_e}, \tag{81}$$

the theorem with IS and PDIS is thus a direct consequence of Proposition 1 in (Angelopoulos et al., 2023).

For WIS, by (Powell & Swann, 1966),

$$\mathbb{E}_{\pi_b}[\tilde{J}_{\text{WIS}}(\tau_i)] = \mathbb{E}^{\pi_b}\left[n\frac{\prod_{t=1}^H \frac{\pi_e(a_t^i|s_t^i)}{\pi_b(a_t^i|s_t^i)}}{\sum_{i=1}^n \prod_{t=1}^H \frac{\pi_e(a_t^i|s_t^i)}{\pi_b(a_t^i|s_t^i)}}J(\tau_i)\right] = V^{\pi_e} + O(\frac{1}{n}). \tag{82}$$

We still have

$$\tilde{J}_{\text{WIS}}(\tau_i) - V^{\pi_e} = \tilde{J}_{\text{WIS}}(\tau_i) - \mathbb{E}_{\pi_b}[\tilde{J}_{\text{WIS}}(\tau_i)] + \mathbb{E}_{\pi_b}[\tilde{J}_{\text{WIS}}(\tau_i)] - V^{\pi_e} = O_p(\frac{1}{\sqrt{n}}) + O(\frac{1}{n}) = O_p(\frac{1}{\sqrt{n}}),$$

$$\tag{83}$$

which indicates the desired result following the standard proof of Proposition 1 in (Angelopoulos et al., 2023).

$\square$

