# OpenReview forum: "PERRY: Policy Evaluation with Confidence Intervals using Auxiliary Data"
_ICLR.cc/2026/Conference — Submitted to ICLR 2026_

### Official Review · Reviewer_uFsk · 2025-10-27

**Soundness:** 2
**Presentation:** 2
**Contribution:** 3
**Rating:** 4
**Confidence:** 3

**Summary:**

The paper proposes two methods that use auxiliary data to construct confidence intervals for off-policy evaluation: CP-Gen and DR-PPI. The authors provide theoretical guarantees to demonstrate that both methods provide valid confidence intervals with a high probability. They validate their approaches empirically across four domains, comparing their approaches to baseline methods including importance sampling.

**Strengths:**

1. The paper studies a significant problem in OPE. Providing reliable CIs for policies with biased data is important and meaningful in many real world domains.

2. The paper provides solid theoretical foundations for their methods under a few assumptions.

3. The empirical comparisons across multiple domains with basline approaches show the effectiveness of their approaches.

**Weaknesses:**

1. Three assumptions in the theory might be strong and the authors do not seem to provide convincing arguments. Specifically, I'm not sure how the fact that policy probabilities lie in [0,1] guarantees Assumption 1 (Line 335). The authors also acknowledge that Assumption 3 is a strong assumption.

2. The CP-Gen algorithm introduces two hyperparameters $\epsilon_s$ and $\epsilon_r$, which are not standard in OPE. The authors do not provide a principled selection rule for these hyperparameters.

**Questions:**

1. How do I understand Table 2? I find the comparison unfair since different methods have different length of CI. It's trivial that DM and $V^{\pi_e}$ have high probability to be biased since they only provide a number; while DR-PPI have much larger CIs. A fair comparison should set the interval length fixed and compare the probability.

2. Can you get probability bounds for your baseline methods? I will evaluate your paper to be much stronger if you can discuss the tradeoff of CI length and valid probability, and show your methods achieve better tradeoffs than baselines.

---

> ### Author Response · Authors · 2025-11-22
>
> We thank the reviewer for their comments and are pleased to hear that our work presents an important and meaningful contribution. We respond to individual reviewer concerns below.
>
> **Assumptions**
>
> We include these theoretical assumptions to enable coverage guarantees for CP-Gen under $\epsilon$-approximation. Importantly, we emphasize that our assumptions still encompass a broad class of real-world MDPs. We will add a short paragraph discussion of this to the main paper for Assumptions 3, based on our explanation below. We also note that the justification for Assumption 1 should be altered, and it is not based on the fact that the policy probabilities lie in [0, 1].
>
> We use Assumption 3 to account for potential errors introduced by the $\epsilon$-approximation used to handle large / continuous state spaces, in which we do not expect to observe many trajectories from identical states and action sequences. We note that in domains with a large number of samples, where we can use a more fine-grained $\epsilon_r$, the Lipschitz assumption here (which is multiplied by $\epsilon_r$ in our theoretical bound) will have much less impact. We have added these clarifications in Section 4.1.
>
> **Hyperparameters**
>
> We note in Section 3.3 that we can set $\epsilon_r, \epsilon_s$ via cross-validation. In particular, we can split the behavior dataset into training and validation sets, then choose the $\epsilon$ that yields the most accurate estimate of the value function $V^{\pi_b}$ on the validation set.
>
> We can also view selecting the correct $\epsilon$ as an optimization problem.  In particular, we note that Theorem 1 presents an upper-bound on $\Delta_w$ that is a function of $\epsilon$. As a result, assuming that we approximate the Lipschitz constants, we can find the optimal $\epsilon$ that results in the lowest value of $\Delta_w$. We have added a discussion of this in Section 3.3.
>
> **Table 2**
>
> In Table 2, we report the ground truth value of the policy $V^{\pi_e}$, intervals for all of the baselines, and a point estimate for DM, which is a method that does not inherently produce an interval. All of our results are reported for a fixed coverage rate of 95% ($\alpha=0.05$) and compare the lengths of the intervals. It is standard to compare interval lengths at a fixed confidence level[1]. In particular, we emphasize that we either have smaller or comparable interval lengths to the best performing method. We have updated the caption associated with Table 2 to reflect this.
>
> **Probability Bounds**
>
> As discussed in the prior comment, we demonstrate that our methods provide either shorter or comparable interval lengths to baseline approaches. In particular, at the same fixed coverage rate, we achieve better interval sizes.
>
> [1] Hanna, J., Stone, P., & Niekum, S. (2017, February). Bootstrapping with models: Confidence intervals for off-policy evaluation. In Proceedings of the AAAI Conference on Artificial Intelligence (Vol. 31, No. 1).

---

> > ### Comment · Reviewer_uFsk · 2025-11-22
> >
> > Thank you for your responses. While my concerns over assumptions are not fully resolved, now I have better understanding of theoretical results and parameter selections. Therefore, I adjust my score accordingly.

---

> > > ### Author Response · Authors · 2025-11-22
> > >
> > > We thank the reviewer for raising their score and are pleased to hear that our clarifications were helpful. We are happy to elaborate on our discussion of the assumptions if there are any particular questions. We emphasize that our assumptions still encompass a broad class of real-world MDPs, including the electronic health records setting explored in our paper.

---

### Official Review · Reviewer_RvJr · 2025-10-30

**Soundness:** 2
**Presentation:** 1
**Contribution:** 2
**Rating:** 4
**Confidence:** 2

**Summary:**

This paper presents a method for off-policy evaluation that can ensure confidence intervals on high-dimensional and long horizon domains, as opposed to prior works that focus on empirical results or bandits settings. Specifically, the paper develops a method for constructing confidence intervals in MDPs with constant initial state and show that a value function can be bounded. Additionally, they propose a method based on doubly robust estimation that can estimate confidence bounds on value over many intiial states. The methods are tested on domains such as robotics, health care, and inventory management.

**Strengths:**

This paper is well motivated and provides novel confidence intervals for OPE (CP-Gen) using an initial starting state and leverages common modern techniques for generative modeling. This is well motivated by applications in healthcare where each decision point at similar states leads to similar behavior. Additionally, the authors relax this assumption in the DR-PPI method for confidence interval calculation by removing the conditioning on starting state.

The authors provide theoretical results that take into account practical considerations and test their method on a wide array of domains (health care, mujoco, robotics, inventory control)

In general, the results demonstrate that the DR-PPI and CP-Gen methods provide the tightest and most accurate bounds on the value estimate of the evaluation policy.

**Weaknesses:**

The main weaknesses in this paper surround the presentation of the methods and results. I struggled to understand the paper fully due to un-named variables, shift in notations, and general lack of clarity / assumptions / givens.

### Lack of clarity
1. What is the purpose of having a discount and a finite horizon?
2. Line 97 introduces IPS with $\pi(s,a)$ but $\pi(a | s)$ is used throughout the rest of the paper. Is there a meaningful difference?
3. Do you have access to the evaluation policy itself or just trajectories thereof?
4. I believe the notation in line 151 of the value function non-standard and it is not clear what $\tau \sim p^{\pi_e}$ means. Is it trajectories sampled from the evaluation policy? Then does $p^{\pi_e}(\tau)$ indicate the probability of $\tau$ under the policy $\pi_e$?
5. In line 162, the authors mention that $\tilde{p}$ is a generative model but give no other context.
6. Line 211, the authors measure the IPS weight for a sample and use $\pi_e$ and $\pi_b$. As mentioned above, I am unclear whether the method has access to the policies themselves or just trajectories.
7. Line 214, the authors discuss a "ball around the output $\delta_{rr'}$" but do not give further context.
8. Line 221, the authors invoke $Q$ which I believe is a Q-function but never define it.
9. Line 253, there is a $E$ which I suspect is an expectation, except the remainder of the paper uses $\mathbb{E}$.
10. Line 262, $\mathbb{V}$ is used but never introduced or discussed.

# Experiments
While the results indicate the efficacy of the methods, there does not seem to evaluation a wide array of evaluation policies. I am curious to see how the performance changes as the distance between the eval and behavior policies grows. The experimentation in varied domains is appreciated, but In general, it would be valuable to see ablations in each environment and discuss the specifics of the behavior and eval polciies.

Nitpick: PERRY is not mentioned anywhere but title

**Questions:**

I believe my questions are evident from the weaknesses section. Please refer there to clarify the understanding of the paper.

---

> ### Author Response · Authors · 2025-11-22
>
> We thank the reviewer for their comments. We are thrilled to hear that our work is well motivated and that our work takes a meaningful step in the direction of more widely applicable OPE estimators. We respond to individual concerns below.
>
> **Notation**
>
> We apologize for the confusion on notation. We clarify these points below in order of the question, and have revised the paper accordingly.
>
> 2. $\pi(a|s)$ is equivalent to $\pi(s, a)$. We have updated Section 2.2 to reflect this.
>
> 4. The notation for the value function is derived from standard notation in prior work prior work [1]. We have added a step to clarify this. $\tau \sim p^{\pi_e}$ indicates that the trajectory $\tau$ was sampled from the probability distribution associated with $\pi_e$, the target policy. These updates are in Section 3.1.
>
> 5. We are happy to clarify the probability distribution associated with the generative model. In particular, we assume access to some generative model $f$, but make no assumptions about it. $\tilde{p}$ is the transition dynamics associated with the generative model.
>
> 7. The ball around the output is a way for us to describe the closeness of the variables. We add a clarification of this in Section 3.1.
>
> 8. $Q$ here is a quantile.
>
> 9. We apologize, this is a typo. It should be $\mathbb{E}$, which refers to the expectation. We have updated this in Section 3.2.
>
> 10. $\mathbb{V}$ refers to the variance.
>
> [1] Dudík, M., Langford, J., & Li, L. (2011). Doubly Robust Policy Evaluation and Learning. International Conference on Machine Learning.
>
> **Clarification questions about OPE setup**
>
> In OPE, we assume that we have access to the target and behavior policies, and trajectories from the behavior policy, as mentioned in Section 2.2. We assume no access to trajectories from the evaluation policy. This setup has both a discount factor and finite horizon; the discount factor incentivizes more immediate rewards and the finite horizon reflects the data generating process in our environments. We clarify this in Section 2.2.
>
> **Further evaluation of policies**
>
> We appreciate the suggestion to evaluate further combinations of behavior and target policies. We note that we present results across four different environments, each of which has a separate combination of behavior and target policy. In Appendix A, we also evaluate how different levels of misspecification of the behavior policy affect the OPE estimate. If there are additional ablations the reviewer was thinking of, we welcome additional suggestions.

---

### Official Review · Reviewer_wDpu · 2025-10-31

**Soundness:** 1
**Presentation:** 2
**Contribution:** 2
**Rating:** 2
**Confidence:** 5

**Summary:**

This paper studies the problem of uncertainty quantification in a version of the off-policy evaluation setting where the behavior data is additionally augmented by synthetically generated data. The additional synthetic data can be used for OPE estimation, however, the source of the synthetically generated data is expected to have imperfections. The paper proposes methods for accounting for those imperfections in the synthetic data by quantifying the uncertainty of the prediction in the form of a *range* of the off-policy performance estimates, i.e., confidence intervals in which the true value lies with a high probability.

**Strengths:**

The problem of using synthetically generated data for OPE is an important one, especially at a time when interaction data with a decision process can be generated in abundance.

**Weaknesses:**

- Mistake in the proof of Equation (4): Equation (24) → (25), the probability of a trajectory includes the transition probabilities along with the policy probabilities. In typical importance weighting, since the transition function is held constant, those terms cancel out. In this setup considered by this work, since the synthetic data comes from a *different* generative model (with a correspondingly different transition function) those terms cannot cancel out as done in these steps.
  - This correction term is central to all the results in this paper. It is not clear how the results that follow hold.
- The aforementioned error must also affect the proof of the validity of the confidence intervals (Theorem 1).  However, no explicit proof is provided for this theorem, other than a statement that says “this is a direct consequence of the Coupling Lemma”. The coupling lemma hasn’t been stated or referenced.
- If this were to be updated/corrected, practically, the application of this method would require access to the transition probabilities of the generative model. This may often be infeasible in practice, as it amounts to having access to the logits of generative model.
- The motivation of this paper is to use *auxiliary* data. However, in both the methods proposed the generative model is being learnt from a split of the behaviour data itself, which is then used to generate additional data. This is the primary distinction from prior work [1]. This is mechanistically bootstrapping from existing data, rather than including additional data beyond what was available.
- The paper would require a significant re-write. A few examples that stand out:
  - Notation is introduced without being defined beforehand: for example, Equation (6), Q is undefined. Only after referring to [1] it is understood that it may probably refer to quantiles. Expressions in the paper bear a striking resemblance to those in [1].
  - Under Section 2, the second term $\widehat{V}^{\pi_e}_{DR-PPI:2}$ is never defined throughout the paper. Why a generative model is being used for this IS estimate term [L276] is also unclear.
  - Notation overloading: Section 2.1 defines $R$ as the reward function, Section 3.1 uses $R$ to denote return.

---
[1] Foffano, Daniele, Alessio Russo, and Alexandre Proutiere. "Conformal off-policy evaluation in markov decision processes." 2023 62nd IEEE Conference on Decision and Control (CDC). IEEE, 2023.

**Questions:**

Assuming transition probabilities of the generative model for the synthetic data were accessible and used in the computation of the score, what does Assumption 3 (Score Smoothness) get interpreted as? The likelihood ratio of the generated data for a given reward difference is Lipschitz? Is that then a statement about how “off-policy+off-dynamics” the data can be to be meaningfully used for OPE?

---

> ### Author Response · Authors · 2025-11-22
>
> We thank the reviewer for their comments. We are glad the reviewer believes we tackle an important problem in OPE. We respond to individual concerns below.
>
> **Proof Validity**
>
> We would like to correct a critical misunderstanding. We completely agree that we cannot cancel trajectories from two different transition models. Fortunately we do not do this– the step from Line 24 to 25 as mentioned by the reviewer does not cancel out terms that use different transition dynamics. We have added additional proof steps to make this clearer (Appendix Section F.1). We reproduce them here for clarity, starting from Equation 24:
>
> Equation 24: $\mathbb{E}_{\tau \sim p^{\pi_b}, \tilde{\tau} \sim \tilde{p}^{\pi_b} \mid S=s, \Delta\_{rr'}=\delta\_{rr'}}[\frac{P(\delta\_{rr'}|s, \tau, \tilde{\tau})P^{\pi_e}(\tau|s)\tilde{P}^{\pi_e}(\tilde{\tau}|s)}{P(\delta\_{rr'}|s, \tau, \tilde{\tau})P^{\pi_b}(\tau|s)\tilde{P}^{\pi_b}(\tilde{\tau}|s)}]$
>
> Updated Equation 25: $\mathbb{E}_{\tau \sim p^{\pi_b}, \tilde{\tau} \sim \tilde{p}^{\pi_b}|S=s, \Delta\_{rr'}=\delta\_{rr'}}[\frac{P^{\pi_e}(\tau|s)\tilde{P}^{\pi_e}(\tilde{\tau}|s)}{P^{\pi_b}(\tau|s)\tilde{P}^{\pi_b}(\tilde{\tau}|s)}]$
>
> Updated Equation 26: $\mathbb{E}_{\tau \sim p^{\pi_b}, \tilde{\tau} \sim \tilde{p}^{\pi_b}|S=s, \Delta\_{rr'}=\delta\_{rr'}}[\frac{\prod\_{t=1}^H \pi_e(a_t|s_t) p(s\_{t+1}|s\_{t},a_t)\pi_e(\tilde{a}_t|\tilde{s}_t)\tilde{p}(\tilde{s}\_{t+1}|\tilde{s}\_{t},\tilde{a}_t)}{\prod\_{t=1}^H \pi_b(a_t|s_t)p(s\_{t+1}|s\_{t},a_t) \pi_b(\tilde{a}_t|\tilde{s}_t)\tilde{p}(\tilde{s}\_{t+1}|\tilde{s}\_{t},\tilde{a}_t)}]$
>
>
>
>
>
> Updated Equation 27: $\mathbb{E}_{\tau \sim p^{\pi_b}, \tilde{\tau} \sim \tilde{p}^{\pi_b}|S=s, \Delta\_{rr'}=\delta\_{rr'}}[\frac{\prod\_{t=1}^H \pi_e(a_t|s_t)\pi_e(\tilde{a}_t|\tilde{s}_t)}{\prod\_{t=1}^H \pi_b(a_t|s_t)\pi_b(\tilde{a}_t|\tilde{s}_t)}]$
>
>
>
>
>
>
> **Coupling Lemma**
>
> We apologize for the confusion. The coupling lemma is a prior known result from Markov Chain theory. We have clarified this in the updated manuscript (Appendix F.3).
>
> **Requirement for transition dynamics**
>
> We agree that requiring access to the transition dynamics of the model would be difficult in practice. In our work, we do not require access to them. In particular, we sample from the generative model which implicitly samples from the learned transition dynamics. We never assume access to the full transition dynamics.
>
> **Auxiliary Dataset**
>
> The generative model expands the effective support of the observed dataset. While the model is trained on behavior data, it does not merely replay or resample trajectories. Instead, it learns a smooth approximation of the underlying behavior distribution. This allows it to generate auxiliary samples that contain novel combinations of states, actions, and transitions that are not explicitly present in the original dataset, thereby providing additional coverage of the state–action space.
>
> Furthermore, if a pre-trained model is already available for the domain, we can use that model to generate trajectories rather than splitting the data (as mentioned in Section 3.3). We emphasize that our method does not rely on the source of the generated data.
>
> **Notation**
>
> We apologize for the confusion. Q does indeed refer to a quantile, similar in presentation to Foffano et. al. We maintain consistency in the notation from the Foffano paper for clarity. $\hat{V}_{DR-PPI:2}$ is the second cross-validated split of the estimator (as defined in Section 3.2). We also want to clarify that we do not use a generative model to do the correction, and that we use a term that resembles a DR estimator to do the correction. Our correction uses both real and generated trajectories. We are happy to update the reward function notation. In particular, we can use $R$ as the reward function and $J$ to refer to the return. We have updated the manuscript to reflect these clarifications in Section 3.2.

---

> > ### Comment · Reviewer_wDpu · 2025-11-25
> >
> > Thank you for the response. Some unclear points still remain, in addition to the ones not addressed in the reponse.
> >
> > > Proof validity
> >
> > It is clear how the updated equations (25, 26) follow. There is still something wrong about the derivation. Equation (23) has joint distributions over $S, \Delta_{r,r'}, \tau, \tau'$ under $\pi_e$ and $\pi_b$ in the numerator and denominator respectively.
> >
> > In factorizing this distribution in Equation (24), the following issues arise:
> >
> > (1) Why are $\tau$ and $\tilde{\tau}$ conditionally independent given $s$, when in Equations (21, 22) there were not?
> >
> > (2) There is a missing term (in both the numerator and the denominator), that of the probability of the *state*, presumably: $P^{\pi_e}(s)$ and $P^{\pi_b}(s)$, for the factorization to be complete. What model, true or synthetic/generative, must these state visitation distributions be compute under? And do they not introduce a dependence on the true or generative model transition probabilities (the primary concern)?

---

> ### Author Response · Authors · 2025-11-26
>
> We thank the reviewer for their response. We respond to individual points below.
>
> **(1)**
>
> We apply conditional probability techniques to separate the joint distribution mentioned.
>
> In particular, from (23) to (24), the probability of rolling out a real trajectory $\tau$ given the initial state $s$ is independent of the rollout of any generated trajectory $\tilde{\tau}$. Namely, $P^{\pi_e}(\tau|s,\tilde{\tau}) = P^{\pi_e}(\tau|s)$, which is the conditional independence that the reviewer mentioned. Thus, (24) holds by factorizing the conditional distribution of $\delta_{rr’}$, $\tau$, $\tilde{\tau}$ and $s$ sequentially.
>
> More intuitively, we can think of $p$ and $\tilde{p}$ as two Markov kernels. Despite the fact that $\tilde{p}$ is trained based on the information in $p$ (using another independent dataset), after training during trajectory rollout, $p$ and $\tilde{p}$ operate independently.
>
> In (21) and (22), we do not factor out $\tau$ and $\tilde{\tau}$ separately because of two reasons. Namely, this independence is not true if we additionally condition on $\delta\_{rr’}$. Once we condition on $\delta\_{rr’}$ which is the trajectory return difference, we impose a constraint on “how far” the real and generated trajectory should be, so the two rollouts are not independent anymore, i.e.,
> $\mathbb{P}^{\pi_b}\_{\tau,\tilde{\tau}|S,\Delta\_{rr'}}(\tau, \tilde{\tau}| s,\delta_\{rr'}) \neq \mathbb{P}^{\pi_b}\_{\tau | S,\Delta\_{rr'}}(\tau|s,\delta\_{rr'})\mathbb{P}^{\pi_b}\_{\tilde{\tau}|S,\Delta\_{rr'}}(\tilde{\tau}| s,\delta\_{rr'}) $. For this reason,  in (24) we factor out $\delta_{rr’}$ first.
>
> Furthermore, we don’t need to factor out $\tau$ and $\tilde{\tau}$ separately. The aim of (21) and (22) is to induce (23), which is the conditional expectation of an importance ratio.
>
> **(2)**
>
> Note that $s$ is the initial state whose distribution is not affected by the policy. Namely, $P^{\pi_e}(s) = P^{\pi_b}(s) = d_0(s)$, where $d_0$ is the initial state distribution as defined in Section 2.1. Thus, in (24) we cancel the two terms out ($P^{\pi_e}(s)$ and $P^{\pi_b}(s)$), which results in no missing terms.
>
> These terms do not introduce a dependence on the true or generative model transition probabilities, because they are initial state distributions.
>
> As for computation, we need do not need a true synthetic/generative model. As mentioned around line 273, “we assume that the initial-state distribution $d_0$ is known (though our results extend to settings in which $d_0$ must be estimated)”. If we were to estimate $d_0$, the estimation would be purely data dependent. The most intuitive way would be to use an independent part of data and use the empirical distribution of the initial state as $d_0$.
>
> We are happy to answer any other clarifying questions if the reviewer has any.

---

> > ### Comment · Reviewer_wDpu · 2025-11-28
> >
> > Thank you for the response, it answer my questions. I will update my score to 4. For some reason, I am unable to edit the original review. I still believe that improving clarity would substantially strengthen the paper.

---

> > > ### Author Response · Authors · 2025-11-28
> > >
> > > We thank the reviewer for raising their score and are glad that our clarifications were helpful. We are happy to elaborate further if there are any remaining questions. We also emphasize that our updated manuscript incorporates revisions made in response to the initial review, and we are willing to refine additional components if recommended.

---

### Meta-Review · Area_Chair_EeME · 2025-12-26

**Summary:**

Reviewers broadly agreed that the paper tackles a timely and important problem in off-policy evaluation: constructing valid confidence intervals when combining real and generated trajectories. The core idea was recognized as meaningful, and reviewers acknowledged that such a capability fills a genuine methodological gap. The major concern of the technical proof correctness was fully addressed during the rebuttal. However, the reviewers all agree that the paper is hard to read, while during the rebuttal, the authors corrected the typos, however this does not improve the readability of this paper.

**Reviewer Concerns:**

1. Correctness of the Core Proof: Fully addressed
2. Requirement for Access to Transition Dynamics: Fully addressed.
3. Overall Presentation and Readability: partially addressed
4. Breadth of Empirical Evaluation: partially addressed

**Reviewer Scores:**

Reviewer wDpu 2->4
Reviewer uFsk 4->4
Reviewer RvJr 4->4

---

### Decision · Program_Chairs · 2026-01-26

Reject